# The DNA glycosylase NEIL2 is protective during SARS-CoV-2 infection

Nisha Tapryal[1,15], Anirban Chakraborty[1,15], Kaushik Saha [2,13], Azharul Islam[1], Lang Pan[3], Koa Hosoki[4], Ibrahim M. Sayed[5,14], Jason M. Duran[6], Joshua Alcantara [7], Vanessa Castillo[7], Courtney Tindle[7], Altaf H. Sarker [8], Maki Wakamiya[9], Victor J. Cardenas[1], Gulshan Sharma[1], Laura E. Crotty Alexander [10], Sanjiv Sur[4], Debashis Sahoo [11,12] ✉, Gourisankar Ghosh [2] ✉, Soumita Das[5,14] ✉, Pradipta Ghosh [7,10] ✉, Istvan Boldogh[3] ✉ & Tapas K. Hazra [1] ✉

SARS-CoV-2 infection-induced aggravation of host innate immune response not only causes tissue damage and multiorgan failure in COVID-19 patients but also induces host genome damage and activates DNA damage response pathways. To test whether the compromised DNA repair capacity of individuals modulates the severity of COVID-19 infection, we analyze DNA repair gene expression in publicly available patient datasets and observe a lower level of the DNA glycosylase NEIL2 in the lungs of severely infected COVID-19 patients. This observation of lower NEIL2 levels is further validated in infected patients, hamsters and ACE2 receptor-expressing human A549 (A549-ACE2) cells. Furthermore, delivery of recombinant NEIL2 in A549-ACE2 cells shows decreased expression of proinflammatory genes and viral E-gene, as well as lowers the yield of viral progeny compared to mock-treated cells. Mechanistically, NEIL2 cooperatively binds to the 5'-UTR of SARS-CoV-2 genomic RNA to block viral protein synthesis. Collectively, these data strongly suggest that the maintenance of basal NEIL2 levels is critical for the protective response of hosts to viral infection and disease.

Coronavirus disease-2019 (COVID-19), caused by the novel severe acute respiratory syndrome coronavirus-2 (SARS-CoV-2), remains a priority of public health concern worldwide. Like other highly pathogenic coronaviruses (CoVs), i.e., SARS-CoV and Middle East Respiratory Syndrome (MERS)-CoV, the pathogenesis of severe COVID-19 is largely attributed to diffused alveolar damage that eventually leads to the onsets of acute respiratory distress syndrome, acute lung injury, and multiorgan failure[1-6]. Recent evidence suggests that the severity of COVID-19 correlates well with a dysregulated and often exacerbated proinflammatory response, also termed as "cytokine storm"[7-10]. Thus, several biologic interventions specifically targeting inflammatory cytokines and related signaling pathways have been clinically evaluated, including IL-6 inhibitors, IL-1 inhibitors, anti-TNF-α agents, corticosteroids, intravenous immunoglobulin, and colchicine[11-16]. While the efficacy and safety of some of these anti-inflammatory agents in COVID-19 patients are still under investigation[17], those that have been clinically evaluated appear to exhibit adverse side effects[10,18-20], making them undesirable treatment options. Hence, it is imperative to better understand the molecular basis of SARS-CoV-2 pathogenesis for identifying novel targets for effective interventions against COVID-19.

Several studies have indicated that oxidative stress contributes to the severe outcome of COVID-19 patients[21-25]. This is because hyperinflammation and oxidative stress together can generate an

A full list of affiliations appears at the end of the paper. ✉e-mail: dsahoo@ucsd.edu; gghosh@ucsd.edu; sodas@ucsd.edu; prghosh@ucsd.edu; sboldogh@utmb.edu; tkhazra@utmb.edu

excessive level of reactive oxygen species (ROS), consequently leading to oxidative damage of various cellular macromolecules, including host genome that is primarily repaired via the base excision repair (BER) pathway[26–28]. Various proteins involved in the BER pathway, such as 8-oxoguanine glycosylase (OGG1), Poly [ADP-ribose] polymerase 1 (PARP1) and DNA polymerase beta (POLB), have been implicated in viral pathogenesis[29–31]. Additionally, several laboratories, including ours, have reported non-canonical roles of BER/single strand break repair (SSBR) proteins, including PARP1, OGG1 and Nei Like DNA Glycosylase 2 (NEIL2) in modulating innate immune response[31–36]. However, the role of BER/SSBR proteins in the pathogenesis of SARS-CoV-2 remains unexplored to date. While analyzing the expression level of BER/SSBR proteins in publicly available transcriptomic databases of SARS-CoV-2 infected patients, we observed that the expression level of NEIL2, an oxidized base-specific DNA glycosylase, is significantly lower within the lungs of patients suffering from severe COVID-19, compared to those of uninfected individuals or even those patients with milder COVID-19 symptoms. In this study, we have investigated the potential role of NEIL2 in mitigating SARS-CoV-2 infection in vitro and provided biochemical evidence of how NEIL2 regulates SARS-CoV-2 infection induced pathogenesis via multiple mechanisms.

## Results

### The low-level expression of NEIL2 in COVID-19 patients correlates with the severity of disease

SARS-CoV-2 infection-induced expression of soluble inflammatory mediators increase influx of inflammatory cells (neutrophils, macrophages, natural killer and T cells) to the site of infection, leading to uncontrolled inflammation, pulmonary endothelial leakage, and impairing lung function[7,8,37]. SARS-CoV-2 infection and host inflammatory responses also generate ROS that are not only signal transducers but are also inducers of host genome damage, thereby triggering a DNA damage response (DDR)[37]. However, in the absence of any report on the mechanistic link between CoV-2 infection and host genome repair, we analyzed publicly available RNA-seq data (database GSE145926) obtained from the bronchoalveolar lavage fluids (BALFs) of individuals suffering from severe and mild COVID-19, along with healthy individuals as controls. Surprisingly, the level of NEIL2, among other BER/SSBR proteins, was found to be significantly lower in severe COVID-19 patients compared to that of the healthy control population (Fig. 1a–d and Supplementary Fig. 1a, b). Such striking findings of the transcriptomic profile in the BALF specimens were validated in three other independent datasets with whole blood transcriptomics (GSE150728, GSE152641 and GSE161777). It was found that downregulation of NEIL2 took place primarily in the monocyte/macrophage lineages (Fig. 1e–g and Supplementary Fig. 1c). Importantly, decreased transcript levels of NEIL2 correlated well with disease severity, including patients that failed to recover from COVID-19 (Figs. 1a–d, g). Comparative expression profiling of uninfected vs. SARS-CoV-2 infected lung epithelial cells (Calu3, GSE147507) also displayed a significant decrease in NEIL2 transcript levels post infection (Fig. 1h). Consistent with decreased levels of transcripts, a significant reduction in expression of NEIL2 protein was also observed in the SARS-CoV-2-infected lungs, particularly in alveolar epithelial cells, compared to healthy controls (Fig. 1i, j) as analyzed by immunohistochemical (IHC) analysis of paraffin-embedded lung specimens of COVID-19 patients.

Blood cells serve as an excellent alternative to the costly and challenging approach of obtaining airway samples from severe COVID-19 patients. Therefore, we analyzed NEIL2 transcript levels in blood cells isolated from COVID-19 patients admitted to the University of Texas Medical Branch (UTMB) hospital by reverse transcriptase real-time quantitative PCR (RT-qPCR), and indeed found a

significantly lower levels of NEIL2 in COVID-19 patients ($n = 20$) compared to that of healthy individuals ($n = 23$) (Fig. 1k, left panel). Notably, the level of NEIL1, another family member of NEIL DNA glycosylase, was increased (Fig. 1k, right panel) probably due to a compensatory host physiological response in an attempt to maintain homeostasis. These results strongly support the correlation between NEIL2 level in blood cells and the severity of CoV-2 infection.

We next investigated the expression of NEIL2 at mRNA and protein levels in SARS-CoV-2-permissive golden Syrian hamsters[38]. Lung specimens were harvested at 5 days post infection (dpi) with SARS-CoV-2 ($1 \times 10^6$ TCID$_{50}$), when weight loss in the animals reached its peak, for the subsequent assessment of the expression of DNA glycosylases. We found that the expression of NEIL2 protein was significantly lower in infected lungs, compared to uninfected controls (Fig. 2a, b). Transcript level of NEIL2 was significantly decreased within the infected lung, while OGG1 mRNA levels were unaltered (Fig. 2c). To further confirm these observations, immunoblots were performed for assessing the expression of several BER proteins in the nuclear extracts of uninfected and SARS-CoV-2 infected lungs of hamsters at 10 dpi. Again, expression of NEIL2 was observed to be significantly downregulated upon SARS-CoV-2 infection in contrast to the expression of other BER proteins, such as OGG1, NEIL1 and AP-endonuclease 1 (APE1) that were largely unchanged (Fig. 2d). We reported earlier that the loss of NEIL2 leads to significant accumulation of oxidative DNA damage in the animal model and cultured cells[33–35]. We thus analyzed DNA damage accumulation in the lungs of SARS-CoV-2-infected vs. uninfected hamsters using long amplicon qPCR (LA-qPCR)[39]. Indeed, SARS-CoV-2-infected animals showed a significant increase in DNA damage accumulation (Fig. 2e), consistent with the decreased level of NEIL2 in those animals[33]. Together, these data suggest a close link between decreased NEIL2 expression and the severe outcome of SARS-CoV-2 infection.

### Correlation of NEIL2 levels and prognosis of COVID-19 based on patient's sex or age

Next, we investigated the prognostic potential of NEIL2 expression for COVID-19 severity, such as the need for intensive care unit (ICU) admission and the use of a mechanical ventilator (MV), in COVID-19 and non-COVID-19 patient populations. Patients requiring ICU or MV had considerably lower levels of NEIL2 than non-ICU or no-MV patients (Fig. 3a, b); however, no such relationship was found for other DNA glycosylases, such as OGG1 and NEIL3 (Supplementary Fig. 2a, b). Moreover, among hospitalized COVID-19 patients, females aged 40 years or less had significantly higher NEIL2 levels (Fig. 3c) which coincided with their shorter duration of hospitalization compared to males in the same age group (Fig. 3d). These findings support the notion that the sex disparity in COVID-19-related severity/deaths puts males at a greater risk, and that this risk markedly increases with age in both sexes[40]. In the case of OGG1/NEIL3, no such age/sex-specific trends were observed (Supplementary Fig. 2c, d). Furthermore, a study involving 100 hospitalized COVID-19 patients showed a significant correlation between higher NEIL2 levels and a shorter duration of hospitalization (Fig. 3e), unlike OGG1 and NEIL3 levels, which showed no correlation to hospital stay (Supplementary Fig. 2e, f). Collectively, these observations again support a strong link between NEIL2 deficiency and COVID-19 severity.

### Effects of recombinant NEIL2 on SARS-CoV-2 -infection induced inflammatory responses

Given that NEIL2 is an anti-inflammatory protein[33,36] and that the expression of NEIL2 is significantly downregulated in patients with severe COVID-19 and in CoV-2-infected hamsters, we hypothesized that recombinant NEIL2 (rNEIL2) could be used as a modality to

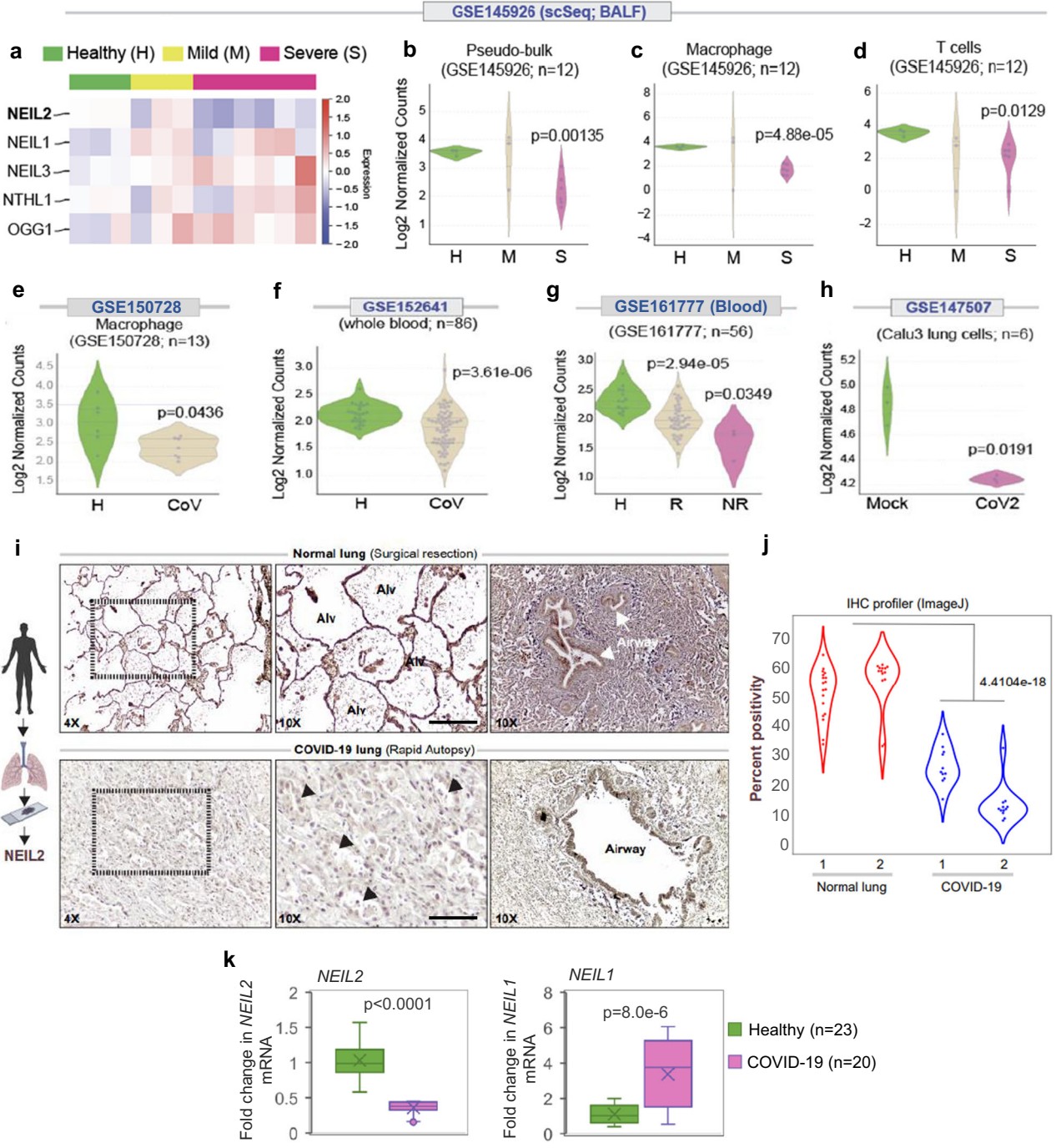

**Fig. 1 | Downregulation of NEIL2 carries a poor prognosis in COVID-19. a** The heatmap displays changes in the expression of BER-associated oxidized base-specific DNA glycosylases in bronchoalveolar lavage fluid (BALF) samples (GSE145926) obtained from healthy controls and patients with COVID-19. Swarm plots display the levels of expression of NEIL2 in Pseudo-bulk (**b**), macrophages (**c**) and T-cells (**d**) as analyzed in healthy (H), mild (M), and severe (S) patients, in the same cohorts as in (**a**). Swarm plots show expression of NEIL2 transcript in macrophages (GSE150728) (**e**) and whole blood (GSE152641) (**f**) from healthy controls (H) and COVID-19 patients (CoV). **g** Swarm plot shows expression of NEIL2 in whole blood (GSE161777) from healthy controls (H) and recovered (R) or not recovered (NR) COVID-19 patients. **h** NEIL2 expression in mock or SARS-CoV-2 infected (CoV2) Calu3 cells (GSE147507). Data was analyzed using log$_2$ (TPM + 1) to compute the final log-reduced expression values for (**b**–**h**) (details in Methods). **i** Schematic

created with BioRender.com shows experimental plan. Normal lung tissue obtained during surgical resection (top) or lung tissue obtained during autopsy studies of COVID-19 patients (bottom) were stained for NEIL2. Representative fields are shown (11-18 fields per sample were imaged and analyzed). Arrowheads = injured alveoli. Alv, alveolar spaces. Scale bar = 200 μm. **j** Violin plots display the intensity of staining between 2 each of healthy normal lungs vs. SARS-CoV-2 infected (COVID-19) lung, as determined by immunohistochemistry (IHC) profiler; data analysis detailed in Methods. **k** Relative expression of *NEIL2* and *NEIL1* transcripts in blood cells of COVID-19 patients ($n = 20$) vs. healthy controls ($n = 23$). Boxes represent the interquartile range extending from first to third quartile, with the horizontal line representing the median value. Whiskers or error bars represent maximum and minimum values; x in box shows the mean. *p*-values (unpaired two-tailed Student's *t* test) vs. healthy controls for k: Source data are provided as a Source Data file.

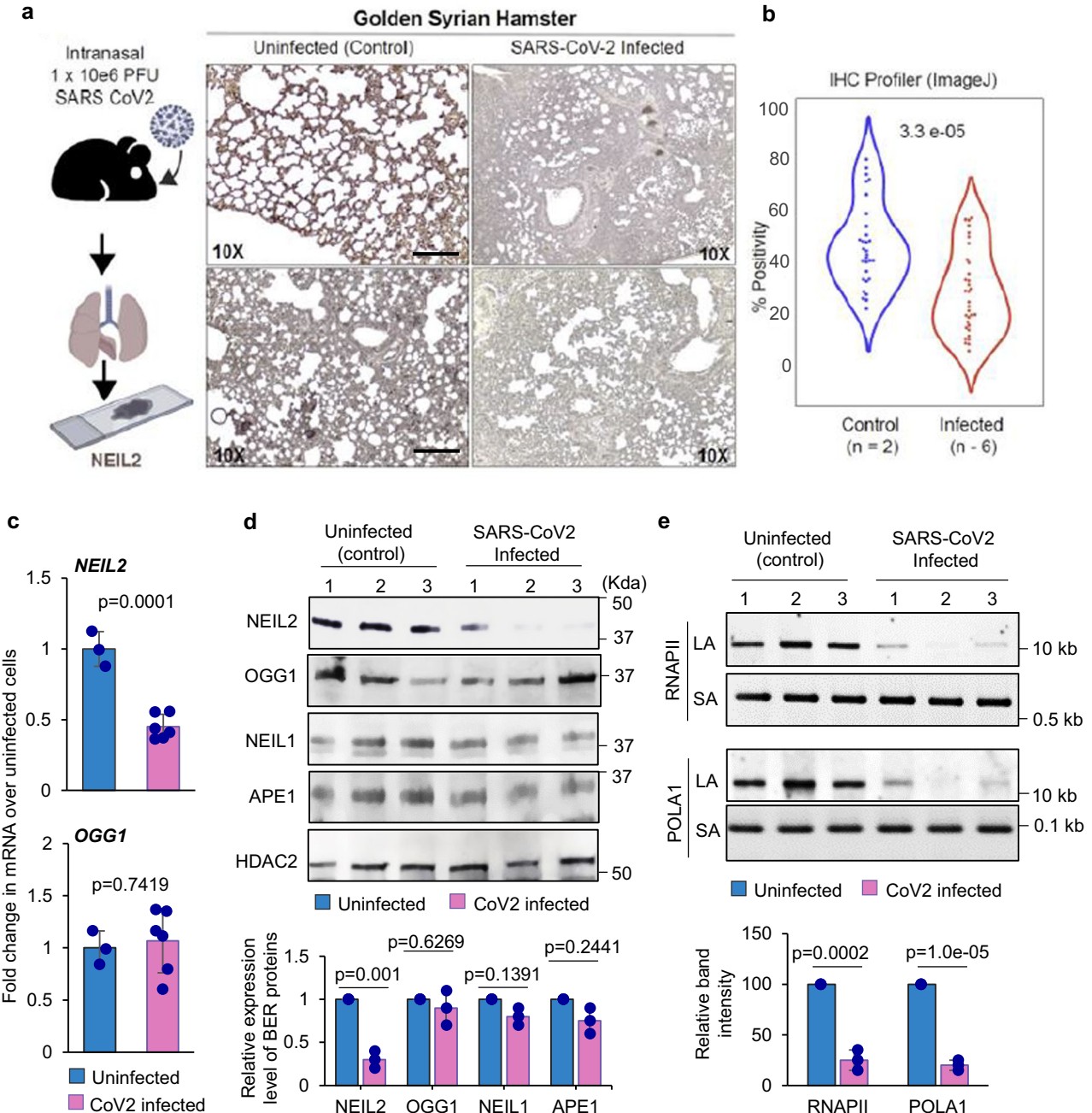

**Fig. 2 | Downregulation of NEIL2 in golden Syrian hamster lungs infected with SARS-CoV-2. a** The schematic created with BioRender.com shows the workflow to generate the tissue sections from hamster lungs post CoV-2 infection (Left Panel). Lungs harvested from hamsters, uninfected (control) or SARS-CoV-2 infected, were analyzed by IHC for NEIL2 expression (Right Panel). Representative images from two different fields are shown. Findings were reproducibly observed in two separate infected cohorts. Scale bar = 200 μm. **b** Violin plots display the intensity of staining for NEIL2, as determined by IHC profiler; data analysis detailed in Methods. **c** Histograms display the levels of expression of *NEIL2* (upper panel) and *OGG1* (lower panel) in hamster lungs collected from infected (*n* = 6 biological replicates) vs. control animals (*n* = 3 biological replicates), as determined by RT-qPCR. **d** Immunoblots from nuclear extracts of control (uninfected) vs. SARS-CoV-2 infected hamster lungs (10 dpi) (*n* = 3 biological replicates) using specific

antibodies against indicated BER proteins (Upper Panel) and quantification of their relative expression level (Lower Panel). HDAC2 was used as a nuclear loading control. **e** Estimation of DNA strand-break accumulation [in two representative genes, RNA polymerase II subunit A (RNAPII) and DNA Polymerase Alpha 1 (POLA1)] in uninfected (control) vs. SARS-CoV-2 infected hamster lungs (10 dpi) (*n* = 3 biological replicates) by LA-qPCR assay (Upper Panel). The normalized relative band intensities are represented in the bar diagram. The mean band intensity of uninfected control mice was arbitrarily set as 100 for (**e**). Representative images are shown from 3 technical replicates with similar results for (**d** and **e**). Error bars represent ± standard deviation from the mean. *p*-values (unpaired two-tailed Student's *t* test) vs. uninfected control for (**c**–**e**): Source data are provided as a Source Data file.

counteract CoV-2-infection and its associated inflammatory responses. Thus, we transduced A549 cells stably expressing human angiotensin converting enzyme-2 (ACE-2), the receptor for the SARS-CoV-2 viral entry[41], with either rNEIL2, rNEIL1, or mock (PBS+carrier) and cells were

then infected with SARS-CoV-2 (WA1-2020 isolate) at the multiplicity of infection (MOI) of 1 for 24 h. Compared to mock-transduced and uninfected cells, we observed comparable NEIL2 mRNA expression in rNEIL2 transduced cells, whereas the NEIL2 transcript level in mock- or

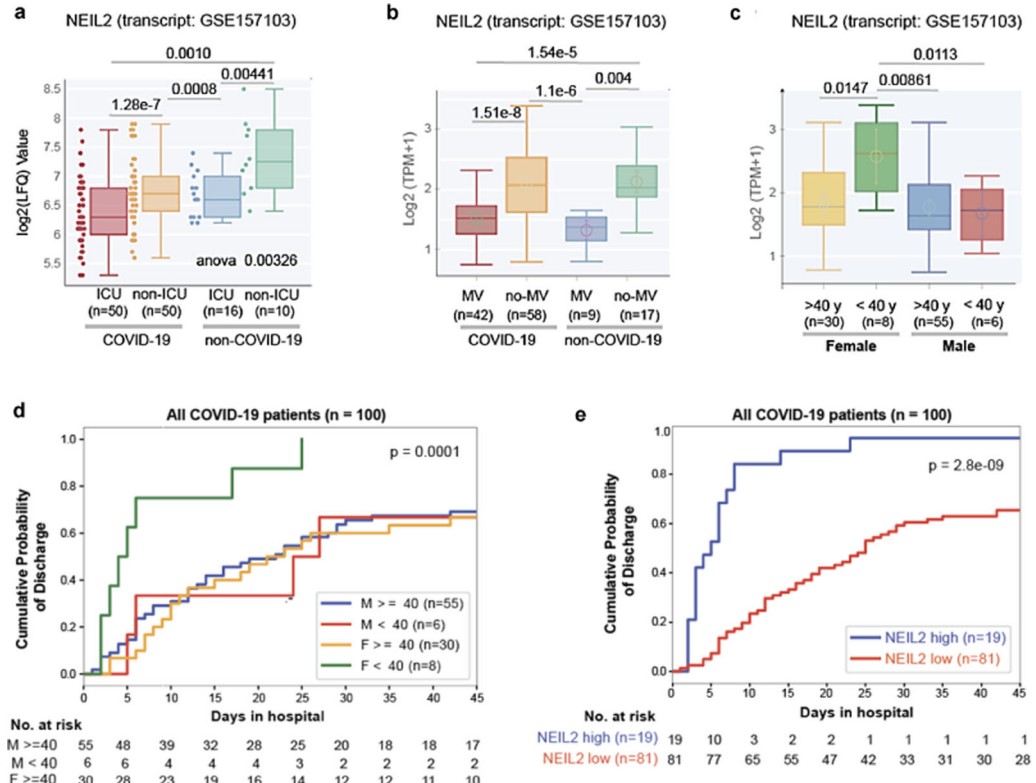

**Fig. 3 | Correlation between NEIL2 levels and severity, and prognosis of COVID-19 based on patient's sex or age.** Boxplots display the levels of expression of NEIL2 in a cohort of hospitalized patients (GSE157103), stratified based on their level of care (Intensive Care Unit [ICU] vs. non-ICU) (**a**) or requirement of Mechanical Ventilator (MV vs. no-MV) (**b**) and diagnosis (COVID-19 vs. non-COVID-19). **c** Whisker plots display the levels of expression of NEIL2 in groups of patients (GSE157103) stratified by sex and age (using 40 years as a cut-off). All boxplots (**a**–**c**) have a horizontal line at the median and the box extends to the first and third quartile with whiskers extending to 1.5-times the interquartile range. Kaplan-Meier plots display the cumulative probability of discharge from the hospital stratified by sex and age (**d**) and high vs. low levels of NEIL2 expression (**e**). Data analysis was performed using lifelines python package version 0.14.6 d (details in Methods). Unpaired two-tailed Student's $t$ test $p$-value to compare two groups was used for statistical analysis.

NEIL1-transduced cells was significantly decreased in response to CoV-2 infection (Fig. 4a), which is consistent with our earlier observation of downregulated NEIL2 expression in CoV-2-infected hamsters and postmortem specimens of COVID-19 patients. However, CoV-2 induced gene expression of *TNFα, IL6, IL1β, CCL2, CCL3* and *CXCL10* was significantly decreased in rNEIL2, compared to mock or rNEIL1 transduced cells (Fig. 4b), confirming the anti-inflammatory function of NEIL2. Surprisingly, we discovered that rNEIL2 transduced cells had significantly lower yield of viral progeny (Fig. 4c) and viral E-gene expression (Fig. 4d), as compared to mock or rNEIL1 transduced cells. Similarly, overexpressing NEIL2 in human gastric adenocarcinoma, AGS cells infected with the human coronavirus 229E strain significantly decreased *IL6* transcript levels (Fig. 4e, upper panel), concomitant with decreased expression of viral E-gene (Fig. 4e, lower panel), when compared with control cells. All these findings strongly suggest that NEIL2 decreases virus-induced inflammatory responses and viral gene expression in CoV-2 infected cells.

## NEIL2 interacts with 5′-and 3′-UTR of SARS-CoV-2 RNA

Like other β-CoVs, SARS-CoV-2 possesses a long RNA genome flanked by 5′- and 3′-untranslated regions (UTRs), containing regulatory cis-acting elements and a stable secondary RNA structure, essential for translation and RNA synthesis[42–44]. Several host proteins interact with the 5′- and 3′-UTRs of viral RNA to either facilitate or hinder viral protein and RNA synthesis[42,44–47]. The decrease in titer of viral progeny and E-gene expression in the presence of rNEIL2 prompted us to test whether NEIL2 is directly involved in the regulation of the viral life cycle via its interaction with regulatory

regions, 5′- and 3′-UTRs of CoV-2 genomic RNA. As SARS-CoV-2, an RNA virus, replicates strictly within the cytoplasm, we first investigated the sub-cellular localization of NEIL2 and consistently found NEIL2 in both nuclear and cytoplasmic compartments of mouse pulmonary cells (Supplementary Fig. 3a). To explore if NEIL2 could interact with UTRs of CoV-2 genomic RNA, we performed RNA chromatin immunoprecipitation (RNA-ChIP) with cell lysates derived from CoV-2 infected A549-ACE2 cells (MOI 1.87) using specific antibodies against NEIL2 or NEIL1 (as control), followed by RT-qPCR using specific primers for CoV-2 5′-and 3′-UTRs or gene body region. We found robust association of NEIL2 with 5′-UTR and 3′-UTR, compared to that of CoV-2 gene body region (Fig. 5a). The UTR-NEIL2 interaction was specific as NEIL1 failed to significantly interact with either the 5′- or 3′-UTR (Fig. 5a). To further confirm the physical interactions between NEIL2 and CoV-2 UTRs, radiolabeled full-length CoV-2 5′-UTR RNA (1 nM) (Sequence in Supplementary Table 2) was incubated with indicated amounts of rNEIL2, and the resulting complexes were analyzed by RNA electrophoretic mobility shift assay (RNA-EMSA) (Fig. 5b). The free probe migrated as two bands, the majority as the fast-migrating natively folded RNA and the minority as the slow migrating unfolded/misfolded RNA[48]. We noted that at 100 nM concentration, rNEIL2 formed a complex represented by a compact band (Fig. 5b, lane 6). Between 50 and 100 nM concentrations, rNEIL2 produced multiple complexes with different migration patterns suggesting that multiple copies of the protein could bind to each 5′-UTR RNA molecules. To further characterize the RNA-NEIL2 protein stoichiometry, we titrated 100 nM unlabeled RNA traced with radiolabeled RNA with rNEIL2. Formation of the compact band was observed with 1.2 μM

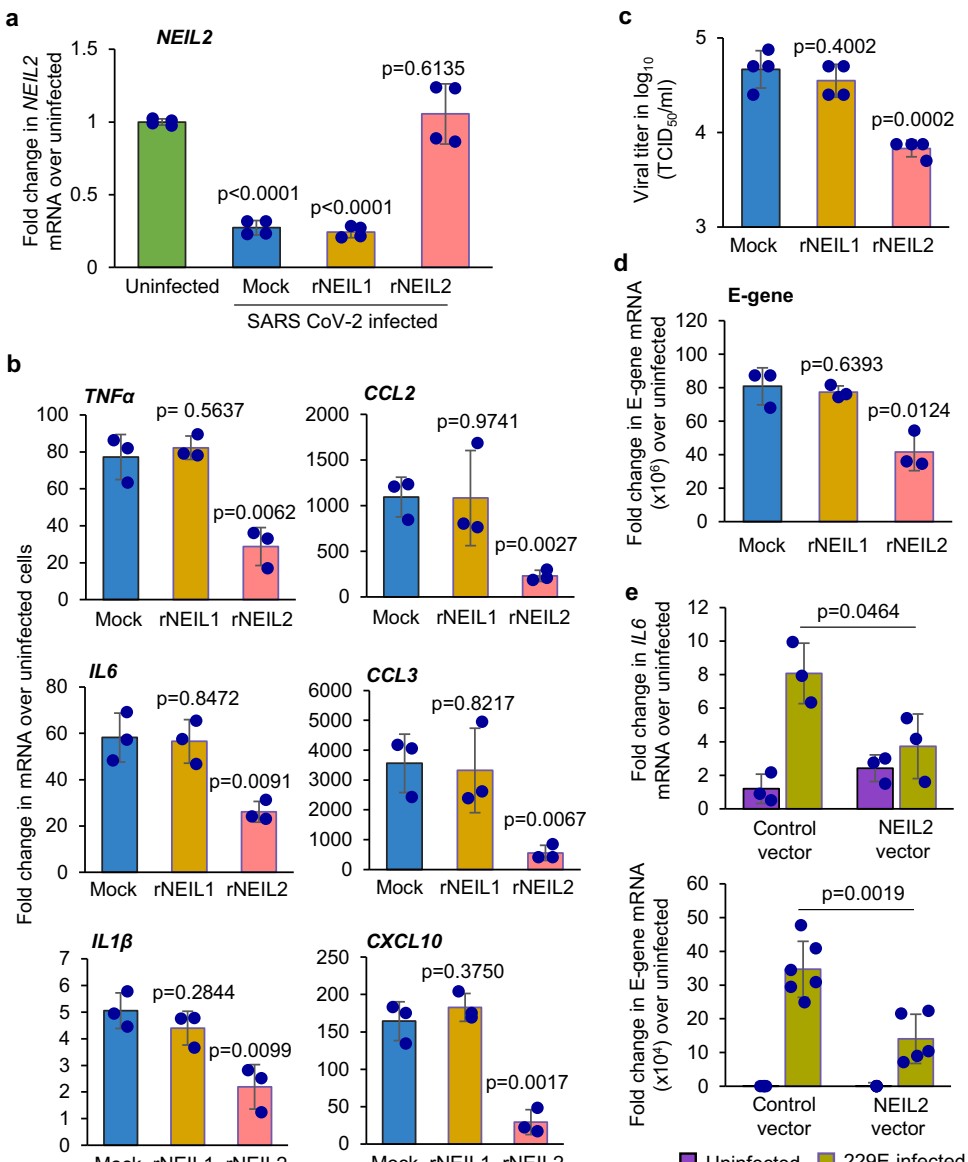

**Fig. 4 | Effect of rNEIL2 on inflammatory gene expression in A549-ACE2 cells following infection with CoV-2. a–d** A549-ACE2 cells were transduced with mock (PBS+ carrier), rNEIL1 or rNEIL2 proteins for 16–24 h, then infected with SARS-CoV-2 (WA1-2020 at MOI 1). Total RNA was isolated at 24 h post infection and expression of *NEIL2* (*n* = 4 biological replicates generated in 3 independent experiments) (**a**) and proinflammatory genes (*n* = 3 independent experiments), as indicated, was analyzed using RT-qPCR (**b**). Supernatants were harvested at 24 h post-infection for viral titer measurement using standard Vero E6 viral titration assay for the supernatants to determine TCID$_{50}$/mL and plotted at log$_{10}$ scale (*n* = 4 biological replicates generated in 3 independent experiments) (**c**). Expression of viral E-gene (n = 3

independent experiments) (**d**) was analyzed by RT-qPCR. **e** AGS cells transfected with control vector (control-vector) or NEIL2 expressing vector (NEIL2 vector) were infected with 229E strain and expression of host *IL6* (*n* = 3 independent experiments) (upper panel) or viral E-gene (*n* = 4–6 biological replicates generated in 3 independent experiments) (lower panel) was analyzed using RT-qPCR compared to uninfected cells 72 h post infection. Target mRNA expression was normalized to *18S* RNA and represented as fold change over uninfected cells. All error bars represent ± standard deviation from the mean. *p*-values (unpaired two-tailed Student's *t* test) vs. uninfected cells for (**a**); mock for **b–d**; and infected control vector expressing cells for (**e**). Source data are provided as a Source Data file.

rNEIL2 (Fig. 5c, lane 10) suggesting that about 12 molecules of rNEIL2 could bind to the 5′-UTR. We subsequently generated a dose-response curve by plotting the percentage of bound RNA along the *y*-axis and log$_{10}$ of rNEIL2 concentrations along the *x*-axis and determined the Hill coefficient to be 2.398 (Fig. 5d). A Hill coefficient greater than 1 indicates that rNEIL2's binding to the 5′-UTR is cooperative. The average K$_d$ for cooperative binding of rNEIL2 to CoV-2 5′-UTR RNA is 44.2 nM. Similarly, 1 nM radiolabeled CoV-2 3′-UTR RNA was titrated with rNEIL2, and the complexes were analyzed by RNA-EMSA (Supplementary Fig. 3b). The free probe primarily migrated as three bands. The majority of the RNA migrated as the natively folded RNA that formed a compact band with 75 nM rNEIL2 (Supplementary Fig. 3b,

lane 5). The fast-migrating RNA species formed the faster migrating RNA-protein complexes (Supplementary Fig. 3b, lanes 4–10, marked by asterisks). Analysis of the unlabeled in vitro transcribed CoV-2 5′- and 3′-UTR RNA on urea gel showed that the 3′-, but not the 5′-UTR RNA forms aggregates (Supplementary Fig. 3c). Thus, it was not possible to accurately measure the free probe intensity in each lane for 3′-UTR because of the background interference caused by upshifting of the faster-migrating RNA species upon binding to rNEIL2 (indicated with asterisks in Supplementary Fig. 3b). To characterize the RNA-protein stoichiometry, we titrated 100 nM unlabeled 3′-UTR RNA traced with radiolabeled RNA with rNEIL2. Formation of the compact band was observed with 1.0 μM rNEIL2 (Supplementary Fig. 3d, lane 9)

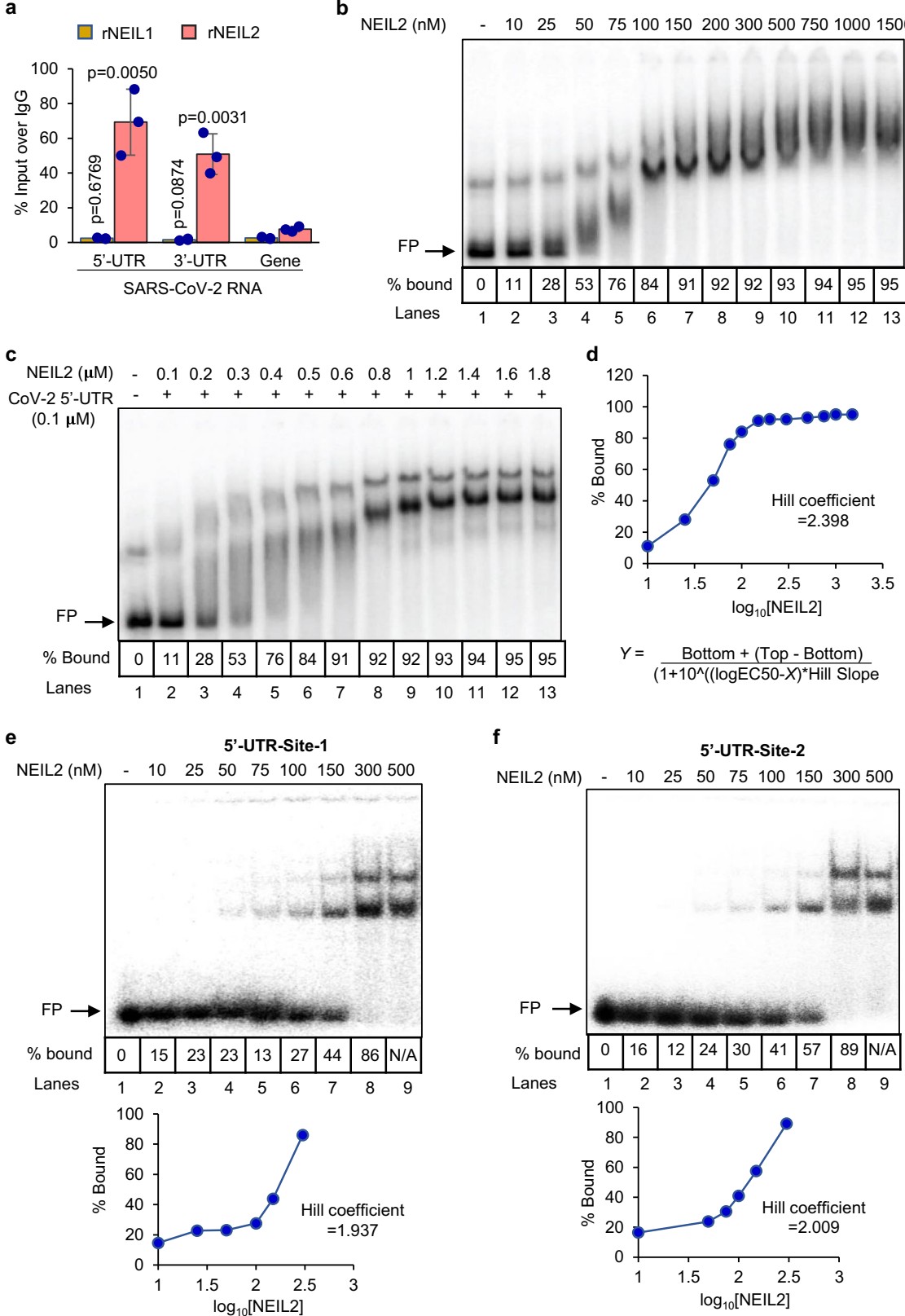

Figure d equation:

$$Y = \frac{Bottom + (Top - Bottom)}{(1+10^{\wedge}((logEC50-X)^* Hill\ Slope)}$$

suggesting that about 10 molecules of rNEIL2 could bind to each molecule of 3′-UTR RNA.

The 5′- and 3′-UTRs of coronavirus RNA contain cis-acting sequences that are functionally important for the binding of viral and host cellular proteins during translation and replication[42,49–51]. Using RNA binding motif search tools (http://www.csbio.sjtu.edu.cn/

bioinf/RBPsuite/, http://rbpmap.technion.ac.il/ and http://cisbp-rna.ccbr.utoronto.ca), we identified two putative zinc finger (ZnF) protein binding sites denoted as 5′-ZnF-site-1 (nt 151-171) and 5′-ZnF-site-2 (nt 215-225) (Supplementary Fig. 4a and Supplementary Table 3) in the 5′-UTR of SARS-CoV-2 genomic RNA. We tested the interaction of rNEIL2 with radiolabeled 38-mer RNA probes containing 5′-ZnF-site-1 or 5′-

**Fig. 5 | Analysis of binding of NEIL2 to the 5′-UTR of SARS-CoV-2. a** A549-ACE2 cells were transfected with rNEIL1 or rNEIL2 proteins for 24 h and then infected with CoV-2 (MOI 1.87). Cell extracts were prepared 8 h post infection and subjected to RNA-ChIP analysis using anti-NEIL1 or anti-NEIL2 antibodies or IgG. RT-qPCR was carried out with 5′-UTR, 3′-UTR or gene body specific primers for CoV-2 RNA. Results are represented as % inputs over IgG where error bars show ± standard deviation from the mean; n = 3 independent experiments. p values (unpaired two-tailed Student's t test) vs. enrichment at gene body. Source data are provided as a Source Data file. **b** Titration of 1 nM radiolabeled CoV-2 5′-UTR RNA probe with rNEIL2 in an affinity-based binding study as analyzed by RNA-EMSA. **c** Titration of 100 nM unlabeled CoV-2 5′-UTR RNA traced with radiolabeled RNA probe with rNEIL2 in a stoichiometry-based binding study as analyzed by RNA-EMSA. **d** Dose-response curve generated by plotting the percentage of bound RNA as shown in b against $Log_{10}$ of rNEIL2 concentrations for calculation of Hill coefficient using the indicated equation, where bottom = 10, top = 94.42 and $EC_{50}$ = 46.17 (top and bottom are plateaus in the unit of the y-axis). Titration of 1 nM radiolabeled CoV-2 5′-UTR- ZnF-site-1 (**e**) and ZnF-site-2 (**f**) containing RNA probes with rNEIL2 in an affinity-based binding study as analyzed by RNA-EMSA. Dose-response curve were generated by plotting the percentage of bound RNA (bottom panels) against $log_{10}$ of rNEIL2 concentrations for calculation of Hill coefficient. FP represents free probe. Representative images and quantitation from n = 3 independent experiments with similar results are shown for (**b**–**f**).

ZnF-site-2 by RNA-EMSA and observed robust sequence specific and dose-dependent binding of the rNEIL2 to both the sites (Supplementary Fig. 4b, site-1, lanes 6-8; site-2, lanes 2–4), but not to the control or mutant RNA oligo (lanes 10-12). Inability of rNEIL1 or zinc finger mutant (C315S) rNEIL2 (ZNFmut-rNEIL2) to bind to 5′-UTR sites of CoV-2 underscored the specificity of rNEIL2-RNA binding (Supplementary Fig. 4c). We also found one ZnF binding site in the 3′-UTR of CoV-2 RNA (3′-UTR-ZnF site, Supplementary Table 3), and indeed, rNEIL2 showed a dose dependent binding to the 3′-UTR-ZnF site, as analyzed by RNA-EMSA (Supplementary Fig. 4d). Furthermore, we titrated CoV-2 5′-UTR ZnF-site-1 and ZnF-site-2 (Fig. 5e, f, respectively), and CoV-2 3′-UTR ZnF-site (Supplementary Fig. 5) containing radiolabeled probes with rNEIL2, and complexes were analyzed by RNA-EMSA. We observed the formation of two RNA-protein complexes with all three RNAs at concentrations of rNEIL2 greater than 50 nM, suggesting 2 molecules of rNEIL2 can bind to short 38-mer RNA probe. Hill coefficients of binding for CoV-2 5′-UTR site-1, site-2, and CoV-2 3′-UTR ZnF-site were estimated to be 1.937, 2.009, and 2.428, respectively. Similar to full length 5′-UTRs, a Hill coefficient greater than 1 suggests cooperative binding for all short RNA probes. The average $K_d$ values for CoV-2 5′-UTR site-1, site-2 and CoV-2 3′-UTR ZnF-site are 176.2, 127.3, and 77.2 nM, respectively. To assess whether such a binding of NEIL2 has any effect on SARS-CoV-2 replication, we examined viral RNA dependent RNA polymerase (RdRp, nonstructural protein, nsp12 in complex with accessory, nsp 7 and nsp 8, Supplementary Fig. 6a) activity in vitro. We used independent RNA oligos containing CoV-2 5′-UTR-ZnF-sites or 3′-UTR-ZnF site sequences as template RNA (sequences in Supplementary Table 3) and short complementary oligo sequence (for 3′-UTR-ZnF-site) as primer to initiate 5′-3′ extension in the presence or absence of rNEIL2. However, we did not detect any NEIL2-mediated inhibition of viral RdRp activity in vitro (Supplementary Fig. 6b, c). Thus, we postulated that NEIL2 regulates viral protein synthesis by blocking activity of host translational machinery at the 5′-UTR of CoV-2.

**NEIL2 suppresses CoV-2 5′-UTR-mediated protein expression**
The data so far strongly suggest that multiple NEIL2 molecules associate with 5′-UTR of CoV-2 genomic RNA and regulates the expression of viral proteins. To address this, we first cloned the 297 nt long 5′-region of CoV-2 genome, containing the full length 5′-UTR sequence, upstream of Green Fluorescence protein (eGFP) in the pcDNA3.1 vector (CoV2-5′-UTR-eGFP, Supplementary Fig. 7a and Supplementary Table 2) and transfected into human lung epithelial BEAS-2B cells, stably expressing c-terminal FLAG-tagged NEIL2 (NEIL2-FLAG). Sixteen hours post transfection, the cell lysates were subjected to RNA-ChIP using anti-FLAG antibody, followed by RT-qPCR analysis. We detected a strong association of NEIL2 with full length SARS-CoV-2 5′-UTR RNA, mimicking the in vivo post infection CoV-2 RNA-NEIL2 interactions (Fig. 5a), but not with the 5′-UTR RNA of several host genes; GAPDH, HPRT and DNA polymerase β (POLB) as controls (Supplementary Fig. 7b). Control reactions without reverse transcriptase ruled out the possibility of DNA contamination in the samples (Supplementary Fig. 7c). Next, we transfected the CoV2-5′-UTR-eGFP plasmid (Supplementary Fig. 7a) or UTR-Less-eGFP plasmid into

NEIL2-FLAG overexpressing or control BEAS-2B cells; and GFP fluorescence was analyzed as a measure of expression of the protein, 12–16 h post transfection. Both CoV2-5′-UTR-eGFP and UTR-less-eGFP plasmid transfected cells showed comparable levels of GFP DNA (as a measure of transfection efficiency) in NEIL2 overexpressing vs. control cells as analyzed by qPCR (Supplementary Fig. 7d). Intriguingly, we observed that the GFP expression was significantly decreased at the protein level (Fig. 6a, left and right panels), but not at the transcript level (Supplementary Fig. 7e) in NEIL2 overexpressing cells compared to control cells, when transfected with the CoV2-5′-UTR-eGFP reporter construct. However, no significant change in GFP expression was observed between control or NEIL2 overexpressing cells transfected with UTR-Less-eGFP plasmid (Fig. 6b, left and right panels). Furthermore, siRNA mediated NEIL2 depletion (Supplementary Fig. 7f) in HEK-293 cells resulted in significantly higher expression of GFP in cells transfected with CoV2-5′-UTR-eGFP construct compared to control siRNA-treated (siControl) cells (Fig. 6c). However, the UTR-less-eGFP plasmid transfected into NEIL2-deficient (siNeil2) cells showed only a modest decrease in GFP expression (Fig. 6d). Finally, we demonstrated that rNEIL2 transfection significantly blocked the expression of CoV-2-Spike glycoprotein in A549-ACE2 cells, in comparison to mock transfected cells, at both 24 and 48 h post CoV-2 infection (Fig. 6e). Collectively, these data suggest that NEIL2 binds to the 5′-UTR of CoV-2 genomic RNA and blocks the host translational machinery, leading to decreased viral protein synthesis.

## Discussion
Dysregulation and often exacerbation of immune responses caused by viral infections could result in severe tissue damage, eventually leading to multiorgan failure and death. The excessive ROS generated as byproducts cause damage to genomic DNA and activate DDR pathways[52,53]. We thus postulated that SARS-CoV-2-infected individuals with compromised DNA repair capacity would be more prone to the onset of severe COVID-19. However, to date, there is no report describing the linkage between SARS-CoV-2 infection and host genome damage-induced signaling, nor the role of DNA repair proteins therein. Here we report that the level of NEIL2, a DNA glycosylase, is significantly lower at both transcript and protein levels in patients suffering from severe COVID-19 and CoV-2 infected golden Syrian hamsters. In addition to the canonical function of repairing genome damage, the work presented here elucidated two non-canonical functions of NEIL2 that can explain its protective role against SARS-CoV-2 infection. We reported earlier that NEIL2 acts as a repressor of NF-κB, a transcriptional activator of proinflammatory genes such as TNFα, IL6 and IL1β[36]. We have also reported recently, that rNEIL2 inhibits IFN-β mediated proinflammatory gene expression by antagonizing NF-κB's access to IFN-β promoter, early after respiratory syncytial virus infection[54]. In the present study, we have demonstrated that transduction of rNEIL2 in A549-ACE2 cells significantly inhibited SARS-CoV-2 induced TNFα, IL6, IL1β, CCL2, CCL3 and CXCL10 expression, further confirming anti-inflammatory role of NEIL2. Therefore, NEIL2 is able to mitigate the viral-induced 'cytokine storm' in the host by acting as a repressor of proinflammatory gene expression. Of note,

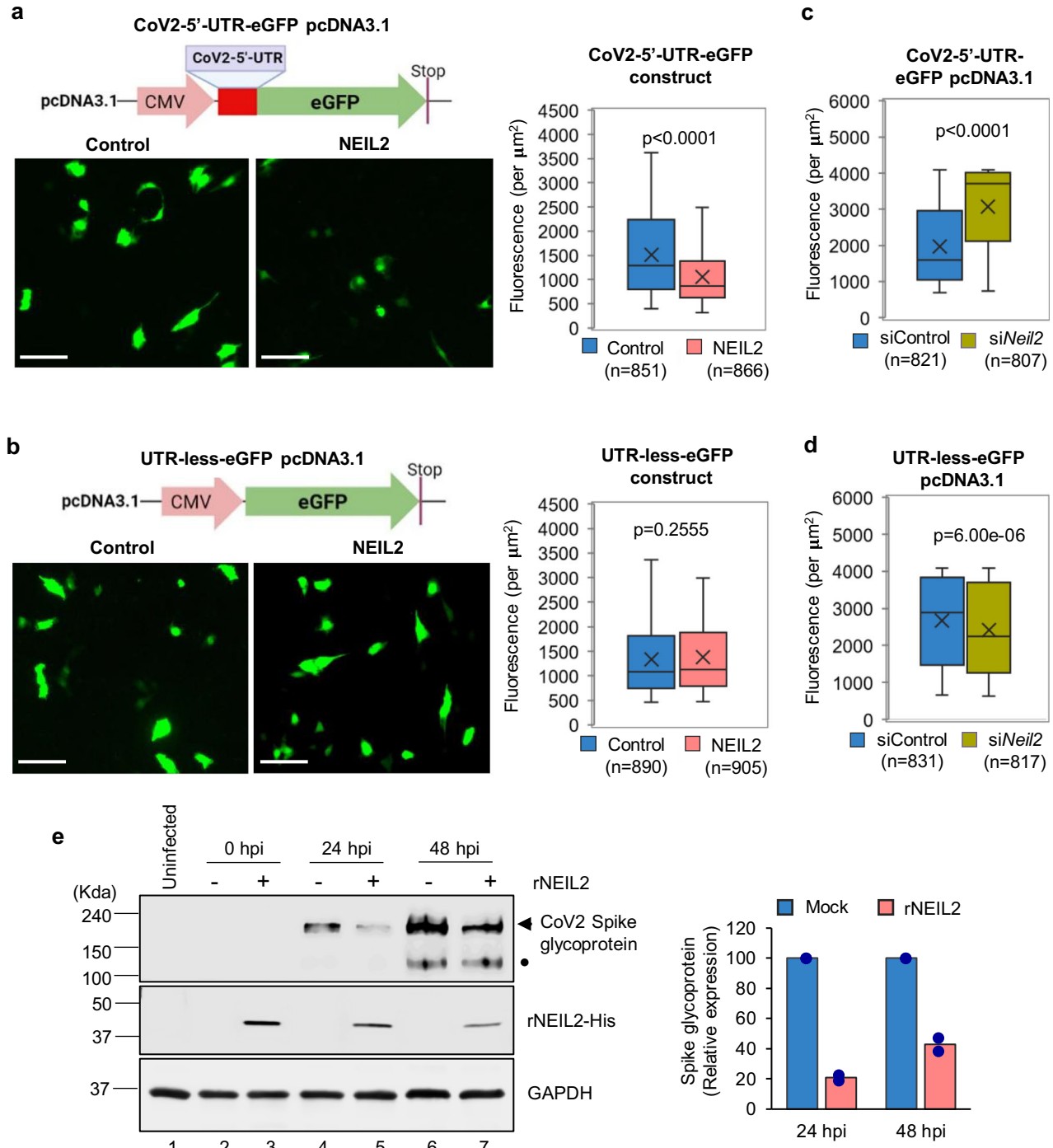

**Fig. 6 | Effect of NEIL2 on SARS-CoV-2 protein synthesis.** Human BEAS-2B cells, control or stably expressing NEIL2-FLAG (NEIL2) were transfected with CoV-2-5'-UTR-eGFP plasmid construct (**a**) or a plasmid devoid of CoV-2-5'-UTR (UTR-Less-eGFP) (**b**), and GFP protein expression was analyzed in live cells after 16 h. Right panels; GFP Fluorescence per µM$^2$ was analyzed from 10 to 14 frames generated in 3 independent experiments. Represented images are shown (Left Panel); Scale bar: 100 µm. HEK293 cells were transfected with control or NEIL2 specific siRNA for 48 h, then transfected with CoV2-5'-UTR-eGFP construct (**c**) or UTR-Less-eGFP plasmid (**d**). GFP fluorescence per µM$^2$ was measured in live cells from 10 to 14 frames generated in 3 independent experiments. All boxplots (**a**–**d**) have a horizontal line at the median and the box extends to the first and third quartile with whiskers extending to 1.5-times the interquartile range; x in the box shows the mean. *n* = number of cells used in data analysis. **e** A549-ACE2 cells were transfected with mock (-) or rNEIL2 (+) for 24 h and then infected with SARS-CoV-2. Cell lysates were prepared at 0, 24 and 48 hpi, and subjected to immunoblot analysis using specific antibodies for CoV-2 spike glycoprotein, His-NEIL2 and GAPDH as indicated. • denotes CoV-2 spike glycoprotein S1 domain. Histogram shows quantification of relative expression level of CoV-2 spike glycoprotein normalized to GAPDH expression (*n* = 2 biological replicates). All *p* = values (unpaired two-tailed Student's *t* test) shown vs. control cells for (**a**–**d**). Source data are provided as a Source Data file.

another oxidized base-specific DNA glycosylase, OGG1 facilitates NF-κB's binding to its consensus sequence and promotes inflammatory response[32,55]. Mechanistically, binding of OGG1 to 8-oxoguanine, located outside of NF-κB motifs, bends DNA sharply producing a kink in the duplex, consequently creating a stereo-specific interface that helps NF-κB in identifying and binding the consensus motif[55]. Conversely, however, NEIL2 keeps NF-κB's activation under control by blocking its binding to consensus motif on target promoter sites[36]. Therefore, an opposing interplay between NEIL2 and OGG1 is essential for unwarranted transactivation of NF-κB and maintaining immune homeostasis. Thus, we conclude that downregulation of NEIL2 would further shift the balance towards exacerbated proinflammatory responses in SARS-CoV-2 induced pathogenesis.

The success of viruses as pathogens depends on their ability to actively reprogram the host's antiviral defense mechanisms. Activation of antiviral innate immune signaling cascades generally begins with recognition of replication intermediates of viruses by intracellular pattern recognition receptors[56,57] or by a set of zinc finger proteins (ZFPs), such as zinc-finger antiviral proteins ZAP, PARP13, monocyte chemoattractant protein 1-induced protein 1 (MCPIP1) and ZCCHC7 that detect viral RNAs and elicit subsequent antiviral responses. In most of the known cases, these ZFPs recruit both the 5′- and 3′- mRNA decay machinery to degrade the target RNAs[58–62]. Suppression of viral replication can occur via translational repression of its own proteins, as has been shown for influenza A virus NS1 mRNA by ZFP 36[60]. ZAP also has an inhibitory effect on the viral translation by disrupting the interaction between eIF4G and eIF4A[63]. Similarly, we provide evidence here that NEIL2, a ZFP[64], also directly interacts with the CoV-2 5′-UTR and blocks viral protein synthesis. This data is in accordance with previous reports showing that host protein impedes viral protein synthesis via binding to 5′- or 3′-UTRs of viral RNA[59,62]. Although the translational initiation mechanism for SARS-CoV-2 RNA is not completely understood, some reports suggest that the translation of SARS-CoV-2 RNA is independent of cap-binding translation factors (eIF4E and eIF4F)[42]. Also, the CoV-2-5′-UTR is rich in GC content and is capable of forming internal ribosome entry sites to recruit host ribosomes for translating its RNA[65]. Since multiple NEIL2 molecules bind to the 5′-UTR, we postulate that such association of NEIL2 will inhibit ribosome entry or interfere with the assembly of translational machinery and inhibit viral protein synthesis. Due to the cooperative nature of the binding, even a slight enhancement in cellular NEIL2 level would greatly augment its binding to the CoV-2-UTRs. In other words, the cooperativity of binding strengthens the correlation between cellular NEIL2 levels and the inhibition of viral translation. It is likely that other cellular anti-viral proteins may work along with NEIL2 forming a stable multiprotein-RNA complex to strengthen the viral translation block. From a clinical perspective, this could be significant since only a modest augmentation of therapeutically induced NEIL2 level would improve the prognosis drastically. This warrants further investigation in the future.

The interplay between the virus and many host factors play critical roles in determining the final outcomes of viral infection. Here, we show that NEIL2 is an important host factor for providing protection against SARS-CoV-2 infection. Based on the work presented here, we propose that if the levels of NEIL2 in hosts are low or the virus is able to significantly diminish the level of NEIL2 beyond a critical threshold, the virus will successfully establish infection in permissive hosts. In support of this notion, we discovered a correlation between NEIL2 levels and differences in age or sex associated with the risk of severe COVID-19. Additionally, lower levels of NEIL2 not only correlated with a longer period of hospitalization but also with higher instances of admission to ICU or the requirement of MVs among patients with severe COVID-19 disease. Moreover, several studies pointed towards the activation of NF-κB-mediated inflammation to explain the importance of age and sex in COVID-19 severity[40,66–68]. The NF-κB pathway induces a proinflammatory phenotype, known as inflamm-aging, in older patients that is associated with increased levels of oxidative stress, thus driving the sustained levels of inflammation and DNA damage leading to cellular DDR, and further expression of proinflammatory genes[66,68]. It is imperative that NEIL2 coordinates with other host factors to mount an elevated defense against viral infection. Future experiments are required to delineate the coordination between NEIL2 and different host factors. Collectively, here we demonstrate multifaceted functions of the host DNA repair enzyme NEIL2, where it unconventionally regulates COVID-19 pathogenesis, by decreasing host inflammatory response, inhibiting SARS-CoV-2 replication, and repairing host genome damage, thereby mitigating disease severity.

## Methods

### Ethics statement
Human Study: The lung specimens from the COVID-19 positive human subjects were collected using autopsy (study was IRB Exempt). All donations to this trial were obtained after telephone consent followed by written email confirmation with next of kin/power of attorney per California state law (no in-person visitation could be allowed into the COVID-19 ICU during the pandemic). The detailed patient characteristics were published elsewhere (PMID: 34127431). For normal lung tissues, lung biopsies were obtained after surgical resection of lungs by cardiothoracic surgeons as before [https://elifesciences.org/articles/66417]. Deidentified lung tissues obtained during surgical resection, which were deemed excess by clinical pathologists, were collected using an approved human research protocol (IRB no. 101590). Blood samples were obtained from UTMB Biorepository of research subjects with a laboratory diagnosis of COVID-19 that consented to participate in the Clinical Characterization Protocol for Severe Emerging Infections (UNMC IRB no. 146-20-FB/UTMB IRB no. 20-0066). The normal healthy subject's blood cell pellets were obtained under UTMB IRB no. 14-0131 and 20-0097.

Animal (Hamster) study: Lung samples from 8-week-old male Syrian hamsters were generated from experiments conducted exactly as in previously published studies (PMID: 32540903). Animal studies were approved and performed in accordance with Scripps Research IACUC Protocol no. 20-0003 and UTMB IACUC Protocol no. 2005060.

### Analysis of RNASeq datasets
Publicly available COVID-19 gene expression databases were downloaded from the National Center for Biotechnology Information (NCBI) Gene Expression Omnibus website (GEO)[69–71]. If the dataset was not normalized, RMA (Robust Multichip Average)[72,73] was used for microarrays and TPM (Transcripts Per Millions)[74,75] was used for RNASeq data for normalization. We used $\log_2 (TPM + 1)$ to compute the final log-reduced expression values for RNASeq data. Accession numbers for these crowd-sourced datasets are provided in the figures and manuscript. Single Cell RNASeq data from GSE145926 was downloaded from GEO in the HDF5 Feature Barcode Matrix Format. The filtered barcode data matrix was processed using Seurat v3 R package[76]. Pseudo bulk analysis of GSE145926 dataset was performed by adding counts from the different cell subtypes and normalized using $\log_2 (CPM + 1)$. All of the above datasets were processed using the Hegemon data analysis framework[77–79].

### Kaplan–Meier (KM) analysis of outcome
Time (duration in hospital) and status (whether the patient is discharged from hospital) were derived from the hospital-free days post 45-day follow-up from COVID-19 patients ($n = 100$, GSE157103). All non-COVID-19 patients ($n = 26$, GSE157103) were excluded from the analysis. Kaplan–Meier (KM) analysis is performed using lifelines python package version 0.14.6. All KM analyses use the *StepMiner* threshold + 0.5 noise margin as the threshold to separate the patients into high and low groups.

## Immunohistochemistry (IHC)

COVID-19 samples were inactivated by storing in 10 % formalin for 2 days and then were transferred to zinc-formalin solution for another 3 days. The decontaminated tissues were transferred to 70% ethanol and cassettes were prepared for tissue sectioning. The slides containing hamster and human lung tissue sections were de-paraffinized in xylene (Sigma-Aldrich, catalog no. 534056) and rehydrated in graded alcohols to water. For NEIL2 antigen retrieval, slides were immersed in Tris-EDTA buffer (pH 9.0) and boiled for 10 min at 100 °C inside a pressure cooker. Endogenous peroxidase activity was blocked by incubation with 3% $H_2O_2$ for 10 min. To block non-specific protein binding 2.5% goat serum (Vector Laboratories, catalog no. MP-7401) was added. Tissues were then incubated with rabbit anti-NEIL2 polyclonal antibody (in house generated, 33) for 1.5 h at room temperature in a humidified chamber and then rinsed with TBS or PBS 3x, 5 min each. Sections were incubated with horse anti-rabbit IgG (Vector Laboratories, catalog no. MP-7401) secondary antibodies for 30 min at room temperature and then washed with TBS or PBS 3x, 5 min each; incubated with 3,3'-diaminobenzidine tetrahydrochloride (DAB) (Thermo Scientific, catalog no. 34002), counterstained with hematoxylin (Sigma-Aldrich, catalog no. MHS1) for 30 s, dehydrated in graded alcohols, cleared in xylene, and cover slipped. Epithelial and stromal components of the lung tissue were identified by staining duplicate slides in parallel with hematoxylin and eosin (Sigma-Aldrich, catalog no. E4009) and visualizing by Leica DM1000 LED (Leica Microsystems, Germany).

## IHC Quantification

IHC images were randomly sampled at different $300 \times 300$ pixel regions of interest (ROI). The ROIs were analyzed using IHC Profiler[80]. IHC Profiler uses a spectral deconvolution method of DAB/hematoxylin color spectra by using optimized optical density vectors of the color deconvolution plugin for proper separation of the DAB color spectra. The histogram of the DAB intensity was divided into 4 zones: high positive (0–60), positive (61–120), low positive (121–180) and negative (181–235). High positive, positive, and low positive percentages were combined to compute the final percentage positive for each ROI. The range of values for the percent positive is compared among different experimental groups.

## Lung tissue specimens from the rapid autopsy procedure

Lung specimens from COVID-19 positive human subjects were collected using autopsy procedures at the University of California San Diego (the study was IRB Exempt) following guidelines from the Centers for Disease Control and Prevention (CDC) and College of American Pathologists autopsy committee. All donations to this trial were obtained after telephone consent followed by written email confirmation with next of kin/power of attorney per California state law (no in-person visitation could be allowed into the COVID-19 ICU during the pandemic). (https://www.cdc.gov/coronavirus/2019-ncov/hcp/guidance-postmortem-specimens.html and https://documents.cap.org/documents/COVID-Autopsy-Statement-05may2020.pdf). Lung specimens were collected in 10 % Zinc-formalin and stored for 72 h before processing for histology as done previously[81,82].

## Human blood samples

Blood cell pellets stored in TRIzol™ LS Reagent (Invitrogen™, catalog no. 10296010) were obtained from the UTMB Biorepository for Severe Emerging Infections from research subjects with a laboratory diagnosis of COVID-19 that consented to participate in the Clinical Characterization Protocol for Severe Emerging Infections (UNMC IRB no. 146-20-FB/UTMB IRB no. 20-0066). Samples were used from subjects categorized as having moderate or severe COVID-19 based on the following criteria: moderate disease if requiring oxygen via nasal cannula, severe disease if requiring oxygen via non-invasive ventilation (e.g., CPAP, BiPAP, High-Flow nasal cannula, venturi mask). The normal healthy subject's blood cell pellets were obtained in TRIzol™ LS Reagent under UTMB IRB # 14-0131 and 20-0097. Total RNA was isolated as per manufacturer's protocol and subjected to real time reverse transcriptase-quantitative Polymerase Chain Reaction.

## Real time reverse transcriptase quantitative polymerase chain reaction (RT-qPCR)

Total RNA extraction was performed from cells using TRIzol™ Reagent (Invitrogen™, catalog no. 15596026) or TRIzol™ LS Reagent. Total RNA (up to 2 µg) was used to synthesize cDNA with a PrimeScript™ RT Kit with gDNA Eraser (TaKaRa, catalog no. RR047A) and qPCR was carried out using TB Green™ Premix Ex Taq™ II (Tli RNase H Plus; TaKaRa, catalog no. RR820A) in Applied Biosystems™ 7500 Real-Time PCR Systems with thermal cycling conditions of 94 °C for 5 min, (94 °C for 10 s, and 60 °C for 1 min) for 40 cycles, and 60 °C for 5 min. The target mRNA levels were normalized to that of *GAPDH* or *18 S* RNA. Primer sequences used in the assay are listed in Supplementary Table 1. In each case, DNase-treated RNA samples without reverse transcriptase were amplified to test genomic DNA contamination.

## Animals

Syrian golden hamsters (Hamster/Golden Syrian Hamster/Male/ 8 weeks old/Charles River/Strain Code 049) experiments were approved by the Scripps Research Institute Institutional Animal Care and Use Committee/Protocol 20-0003, and were carried out in accordance with recommendations. Lung samples were collected from 8-week-old Golden Syrian hamsters post SARS-CoV-2 infection conducted exactly as in a previously published study[38]. Briefly, lungs from hamsters challenged with SARS-CoV-2 ($1 \times 10^6$ PFU) were harvested on day 5 (peak weight loss) and NEIL2 protein and mRNA levels were analyzed by IHC and RT-qPCR, respectively. Syrian golden hamsters (Male/8 weeks old) were infected with SARS-CoV2 as approved by the UTMB IACUC (protocol no. 2005060) and nuclear extract was prepared from the uninfected and infected hamster lungs at 10 days post infection as described before[33,39], and DNA was extracted from the same samples for LA-qPCR.

## Cell culture and transient transfection

A549 cells stably expressing human angiotensin I converting enzyme 2 (A549-ACE2)[83] is maintained in Eagle's Minimum Essential Media (EMEM; Gibco, Cat # 11095080), containing 10% fetal bovine serum (FBS), 100 units/ml penicillin and 100 µg/mL streptomycin. A549-ACE2 cells grown in six-well plates at ~70% confluence were transduced with recombinant proteins using Pierce™ Protein Transfection Reagent according to manufacturer's recommendations (Pierce, Thermo Scientific, catalog no. 89850). In brief, Pierce reagent (dissolved in 250 µL of methanol or chloroform) was evaporated to remove traces of solvent and 2 µg of rNEIL1, or rNEIL2 protein was added in PBS, vortexed, incubated for 5 min at room temperature, then the mixture was supplemented with serum free medium. Mixtures were added directly onto the cell monolayers, incubated for 4 h in a 5% $CO_2$ containing incubator at 37 °C and then one volume of 20% serum-containing medium was added for overnight. Transfection efficiency varied between 68 and 75% as determined in parallel experiments by indirect immunofluorescence assays using anti-NEIL2 or anti-NEIL1 (in house generated[84]) antibodies. Transduced A549/ACE2 cells were infected with SARS-CoV-2 at MOI 1–1.87. After incubation for an hour with viral inoculum, cells were washed three times with EMEM. Infected cells were harvested at indicated time points in various lysis buffers, depending on the downstream experiment. Supernatants from infected cells were harvested at 24 h post-infection for measuring the infectious virus titers by the $TCID_{50}$ assay using Vero E6 cells. Briefly, 50 µL supernatants from infected cells were serially diluted (10-fold) in EMEM supplemented with 2% FBS; 100 µL of serially diluted samples

were added to Vero E6 cells grown in 96-well plates and cultivated at 37 °C for 3 days followed by observation under a microscope for the status of virus-induced formation of cytopathic effect (CPE) in individual wells. The titers were expressed as log TCID$_{50}$/mL.

Human bronchial epithelium cell line, BEAS-2B (ATCC® CRL-9609™) stably expressing NEIL2-FLAG, human gastric adenocarcinoma (AGS, ATCC® CRL-1739™) and human embryonic kidney cells (HEK293[85]) were grown at 37 °C and 5% CO$_2$ in DMEM/F-12 (1:1) containing 10% FBS, 100 units/ml penicillin and 100 units/ml streptomycin. For all experiments, 50–60% confluent cells were used. We routinely tested cell lines for mycoplasma contaminations using the PCR-based Venor™ GeM Mycoplasma Detection Kit (Sigma, catalog no. MP0025). Control or stable BEAS-2B cells at ~70% confluency were transiently transfected with vector expressing GFP with (SARS-CoV2-5′-UTR-eGFP construct, synthesized and cloned by GenScript Inc.) or without (UTR-Less-eGFP construct) UTR (100 ng) using Lipofectamine TM 2000 (Invitrogen, catalog no. 11668027), according to the supplier's protocol. To monitor transfection efficiency, a reporter gene construct (0.25 μg) containing β-galactosidase downstream to the SV40 promoter was co-transfected. Cells were allowed to recover for 16 h in media with serum and then GFP florescence was measured using an ECHO florescent microscope (ECHO Revolve-R4). Total RNA and DNA were isolated for subsequent qPCR analysis.

## Immunoblotting

The proteins in the nuclear extracts (from Hamster lungs)/whole cell extracts A549-ACE2 cells were separated onto a Bio-Rad 4–20% gradient Bis-Tris gel, then electro-transferred on a nitrocellulose (0.45 μm pore size; GE Healthcare) membrane using 1X Bio-Rad transfer buffer. The membranes were blocked with 5% w/v skimmed milk in TBST buffer (1X Tris-Buffered Saline, 0.1% Tween 20) and immunoblotted with appropriate antibodies SARS-CoV-2 spike protein (S1-NTD) (Cell Signaling Technology, catalog no. 56996 S), GAPDH (BioBharati Life Sciences, catalog no. AB0060), Histidine (BioBharati Life Sciences, catalog no. AB0010), NEIL2[33], OGG1 (in-house generated[86]), NEIL1 and APE1 (in-house generated[87]), and HDAC2 (Histone deacetylase 2, GeneTex, catalog no. GTX109642). The membranes were extensively washed with 1% TBST followed by incubation with anti-isotype secondary antibody (Cell Signaling Technology, catalog no. 7074) conjugated with horseradish peroxidase in 5% skimmed milk at room temperature. Subsequently, the membranes were further washed three times (10 min each) in 1% TBST, developed and imaged using kwikquant image analyzer and image analysis software (ver. 5.2) (Kindle Biosciences). Due to cross reactivity of common secondary antibody with the pre developed membrane, the samples were run in parallel gels in similar conditions, and developed with different antibodies. For all the primary antibodies, 1:1000 dilution was used and for secondary antibody, 1:2000 dilution was used.

## RNA Chromatin immunoprecipitation (RNA-ChIP)

RNA-ChIP assays were performed as described earlier[39]. Briefly, cells were cross-linked in 1% formaldehyde for 10 min at room temperature. Then 125 mM Glycine was added for 5 min at room temperature to stop crosslinking and then samples were centrifuged at 1000 g at 4 °C for 5 min to pellet the cells. The cell pellet was re-suspended in sonication buffer, containing 50 mM Tris-HCl pH 8.0, 10 mM EDTA and 1% SDS with 1X Protease inhibitor cocktail and sonicated to an average DNA size of ~300 bp using a sonicator (Qsonica Sonicators). The supernatants were diluted with 15 mM Tris-HCl pH 8.0, 1.0 mM EDTA, 150 mM NaCl, 1% Triton X-100, 0.01% SDS containing protease inhibitors, and incubated with anti-NEIL1, -NEIL2, -FLAG (Millipore, catalog no. F1804) or normal IgG (Santa Cruz, catalog no. sc-2025) antibodies overnight at 4 °C. Immunocomplexes (ICs) were captured by Protein A/G PLUS agarose beads (Santa Cruz, catalog no. sc-2003), that were then washed sequentially in buffer I (20 mM Tris-HCl pH 8.0, 150 mM

NaCl, 1 mM EDTA, 1% Triton-X-100 and 0.1% SDS); buffer II (same as buffer I, except containing 500 mM NaCl); buffer III (1% NP-40, 1% sodium deoxycholate, 10 mM Tris-HCl pH 8.0, 1 mM EDTA), and finally with 1X Tris-EDTA (pH 8.0) buffer at 4 °C for 5 min each. RNase inhibitor (50 U ml$^{-1}$, Roche, catalog no. 03335402001) was added to sonication and IP buffers, and 40 U ml$^{-1}$ to each wash buffer. The ICs were extracted from the beads with elution buffer (1% SDS and 100 mM NaHCO$_3$) and de-crosslinked for 2 h at 65 °C. RNA isolation was carried out in acidic phenol–chloroform followed by ethanol precipitation with GlycoBlue (Life Technologies, catalog no. AM9516) as a carrier. Reverse transcription and cDNA preparation was performed using a PrimeScript RT Kit with gDNA Eraser. RNA-ChIP samples were analyzed by qPCR using specific primers (listed in Supplementary Table 1) and represented as percentage input after normalization to IgG.

## Protein expression and purification

Wild-type recombinant His-tagged -NEIL2, -NEIL2-ZnF mutant (ZnF-NEIL2mut) and -NEIL1 proteins were purified from *E. coli* using protocol as described earlier[64]. Briefly, pET22b (Novagen) vector containing C-terminal 6xHis tagged Coding DNA Sequence (CDS) of various proteins was transformed into *E. coli* BL21(DE3) RIPL Codon-plus cells (Agilent technologies, catalog no. 230280). The log-phase culture (A$_{600}$ = 0.4–0.6) of *E. coli* was induced with 0.5 mM isopropyl-1-thio-β-D-galactopyranoside (IPTG) and grown at 16 °C for 16 h. After centrifugation, the cell pellets were suspended in a lysis buffer (Buffer A) containing 25 mM Tris-HCl, pH 7.5, 500 mM NaCl, 10% glycerol, 1 mM ß-mercaptoethanol (ß-ME), 0.25% Tween 20, 5 mM imidazole, 2 mM phenylmethylsulfonyl fluoride (PMSF). After sonication, the lysates were spun down at 13,000 rpm and the supernatant was loaded onto HisPur™ Cobalt Superflow Agarose (Thermo Scientific™, catalog no. 25228), previously equilibrated with Buffer A, and incubated for 2 h at 4 °C. After washing with Buffer A with a gradient of increasing concentration of imidazole (10, 20, 30, 40 mM), the His-tagged proteins were eluted with an imidazole gradient (80–500 mM imidazole in buffer containing 25 mM Tris-HCl, pH-7.5, 300 mM NaCl, 10% glycerol, 1 mM ß-ME, 0.25% Tween 20). After elution, the peak protein fractions (in the range of 100–250 mM imidazole) were dialyzed against Buffer C (1X PBS, pH 7.5, 1 mM dithiothreitol (DTT), and 25% glycerol) and stored at −20 °C in aliquots.

The Corona virus nsp12 (GenBank: MN908947) gene, cloned into a modified pET24b vector, with the C-terminus possessing a 10 × His-tag, was a gift from Dr. Whitney Yin. The plasmid was transformed into *E. coli* BL21 (DE3) RIPL Codon-plus cells, and the transformed cells were cultured at 37 °C in LB media containing 100 mg/L ampicillin. After the OD$_{600}$ reached 0.8, the culture was cooled to 16 °C and supplemented with 0.5 mM IPTG. After overnight induction, the cells were harvested through centrifugation, and the pellets were re-suspended in lysis buffer (20 mM Tris-HCl, pH 8.0, 150 mM NaCl, 4 mM MgCl2, 10% glycerol). The rest of the procedure is same as above with following modifications: the His-tagged protein was eluted with an imidazole gradient (80–250 mM imidazole in buffer containing 20 mM Tris-HCl, pH 8.0, 150 mM NaCl, 4 mM MgCl$_2$, 10% glycerol). Similarly, nsp7 and nsp8 genes, individually cloned in pET22b and pET30a+ vectors, respectively, were expressed in *E. coli* as described in case of NEIL proteins. After elution, the peak protein fractions of these proteins were dialyzed against Buffer D (20 mM Tris-HCl, pH 8.0, 250 mM NaCl, 1 mM DTT, 25% glycerol) and stored at −20 °C in aliquots.

## RNA dependent RNA polymerase (RdRp) assay

For assembling the stable nsp12-nsp7-nsp8 complex, purified nsp12 was incubated with nsp7 and nsp8 at 4 °C for three hours, at a molar ratio of 1: 2: 2 in a buffer containing 20 mM Tris-HCl, pH 7.5, 250 mM NaCl and 4 mM MgCl$_2$[88].

RdRp assay for CoV-2-5′-UTR ZnF-site was conducted using a self-priming RNA oligo and one short RNA oligo was used as the primer for

such assay for CoV-2-3′-UTR ZnF-site containing sequence as template (Supplementary Table 3). Oligos were mixed at the following final concentrations in 20 μL reaction volume: Tris-HCl (pH 8, 25 mM), RNA short primer (200 μM), RNA template (2 μM), [α-³²P]-UTP (0.1 μM), BSA (1 mg/ml), 0.1 μM GTP, CTP, ATP and 0.01 μM UTP and SARS-CoV-2 RdRp complex (~0.1 μM) on ice. For NEIL2 binding, the indicated concentrations of NEIL2 were incubated in the buffer with RNA on ice for 15 min. Reactions were stopped after 15, 30 or 60 min by the addition of 20 μL of a formamide/EDTA (50 mM) mixture and incubated at 95 °C for 10 min. Samples were run in a 8% urea PAGE using 1x Tris-borate-EDTA as the running buffer. The gels were exposed to a Phosphor screen for 4–6 h and images analyzed using a Typhoon FLA 7000 phosphorimager (GE Healthcare).

### RNA-Electrophoretic mobility-shift assay (RNA-EMSA)

RNA-EMSAs with full length CoV-2 5′- and 3′-UTRs were carried out as previously described[89] with some modifications. Briefly, the 297-nt long 5′- and the 200-nt long 3′-UTR RNAs (sequences in Supplementary Table 2, synthesized and cloned in plasmids by GenScript Inc.) were synthesized by in vitro transcription and end-labeled with [γ-³²P] ATP. The indicated concentrations of components were mixed in 15 μl reactions containing 0.3% poly (vinyl alcohol) (Sigma, catalog no. P-8136), 2 mM MgCl₂, 0.1 U RNase inhibitor (Biobharati Life Science, India), 1 mM DTT, 20 mM HEPES-NaOH pH 7.5, 150 mM NaCl, and 20% glycerol, and incubated at room temperature for 5 min. The RNA-protein complexes were resolved on a native gel (4% 89:1 polyacrylamide gel containing 2.5% glycerol, 50 mM Tris, and 50 mM glycine) at 4 °C for 90 min. Our EMSAs were designed to examine both the affinity (when RNA is in trace amount) and the stoichiometry (when RNA is not in trace amount) of the protein component required to form complexes following principles described before[90]. Hill coefficient was calculated as described before[89]. RNA-EMSA with short (38-mer) oligonucleotide (oligo) probes were performed as described before[36,91], with some modifications. Sequences of the oligonucleotides are listed in Supplementary Table 3. Briefly, [γ-³²P]ATP labeled RNA oligos were incubated with 10–1000 nM of purified protein in a binding buffer containing 10 mM Tris-HCl buffer (pH 7.6), 15 mM KCl, 5 mM MgCl₂, 0.1 mM DTT, 10 U of RNase inhibitor, 1 μg BSA, and 0.2 mg/ml yeast tRNA in a 10–20 μl reaction volume. After a 10-min incubation at room temperature RNA-protein complexes were resolved on a 5% non-denaturing polyacrylamide gel at 120 V using 0.5x Tris-borate-EDTA as the running buffer at 4 °C. For titration assays with short oligos the reaction mix was prepared without yeast tRNA. Gels were fixed in an Acetone: Methanol: H₂O (10:50:40) solution for 10 min, exposed to a Phosphor screen for 12–16 h and scanned using Typhoon FLA 7000 phosphorimager.

### Long Amplicon qPCR (LA-qPCR) assay

Lung tissues from freshly euthanized uninfected and SARS-CoV-2 infected hamsters were used for DNA damage analysis. Genomic DNA was extracted using the Genomic tip 20/G kit (Qiagen) per the manufacturer's protocol, to ensure minimal DNA oxidation during the isolation steps. The DNA was quantitated by Pico Green (Molecular Probes) in a black-bottomed 96-well plate and gene-specific LA-qPCR assays were performed as described earlier[33,39] using LongAmp® Taq DNA Polymerase (New England Biolabs, Catalog no. M0323). The LA-qPCR reaction was set for all genes from the same stock of diluted genomic DNA sample, to avoid variations in PCR amplification during sample preparation. Preliminary optimization of the assays was performed to ensure the linearity of PCR amplification with respect to the number of cycles and DNA concentration (10–15 ng). The final PCR reaction conditions were optimized at 94 °C for 30 s; (94 °C for 30 s, 55–60 °C for 30 s depending on the oligo annealing temperature, 65 °C for 10 min) for 25 cycles; 65 °C for 10 min. Since amplification of a small region is independent of DNA damage, a small DNA fragment (~200–500 bp)

from the corresponding gene(s) was also amplified for normalization of amplification of the large fragment. Primer sequences used in the assay are listed in Supplementary Table 1. The amplified products were then visualized on gels and quantitated with ImageJ software (NIH). The extent of damage was calculated in terms of relative band intensity with the uninfected control mice/hamster sample considered as 100.

### Statistical analysis

All statistical tests were performed using R version 3.2.3 (2015-12-10). Standard t-tests were performed using python scipy.stats.ttest_ind package (version 0.19.0) with Welch's Two Sample t-test (unpaired, unequal variance (equal_var=False), and unequal sample size) parameters. Multiple hypothesis correction was performed by adjusting $p$ values with statsmodels.stats.multitest.multipletests (fdr_bh: Benjamini/Hochberg principles). The results were independently validated with R statistical software (R version 3.6.1; 2019-07-05). Pathway analysis of gene lists were carried out via the Reactome database and algorithm. Reactome identifies signaling and metabolic molecules and organizes their relations into biological pathways and processes. Kaplan-Meier analysis was performed using lifelines python package version 0.14.6. Violin and Swarm plots were created using python seaborn package version 0.10.1.

Graph generation and analysis of statistical significance between two sets of data were performed with Microsoft excel, GraphPad Software (https://www.graphpad.com/quickcalcs/pvalue1.cfm) and MedCalc statistical software (https://www.medcalc.org/calc/comparison_of_means.php). $p$ = values < 0.05 were considered statistically significant.

### Reporting summary

Further information on research design is available in the Nature Portfolio Reporting Summary linked to this article.

## Data availability

All data supporting the conclusions of this study can be found in the Article, Supplementary and Source Data files. Computational data is available at the Github link: https://github.com/sahoo00/BoNE/blob/master/neil2/neil2-091223.ipynb. Source data are provided with this paper.

## Code availability

All computational codes used in this work to enable others to reproducibly generate the panels can be found at https://github.com/sahoo00/BoNE/blob/master/neil2/neil2-091223.ipynb.

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

## Acknowledgements

This work was supported by the National Institutes for Health grants, RO1 NS073976 (to T.K.H.); RO1HL145477 (to S.S. and T.K.H.); R01 DK107585 (to S.D.); R01 AI141630 (to P.G.); R01 GM138385 (to D.S.) and RO1 AI155696 (to P.G., D.S. and S.D.); R01 GM085490 and AI163327 (to G.G.). US National Institute of Allergic and Infectious Diseases, grant no. AIO62885 (to I.B and T.K.H.); US Department of Defense grant no. W81XWH-18-1-0743 (to S.S.). University of California Tobacco Related Disease Research Program (TRDRP) grant (26IR-0017 to A.H.S). The authors would like to thank Dr. Chien-Te K. Tseng and his lab members, Dr. Kempaiah Rayavara, Dr. Pinghan Huang and Mr. Jason C. Hsu (Department of Microbiology and Immunology, The University of Texas Medical Branch (UTMB)), for kindly providing SARS-CoV-2 infected cells. We thank Dr. Junki Maruyama and Dr. Slobodan Paessler (Department of

Pathology, UTMB) for generously providing SARS CoV-2 infected hamster tissue samples. The authors would like to thank Victor Pretorius, Rachel White and Jen Bigbee (Department of Cardiothoracic Surgery, UC San Diego) and the staff of the Department of Pathology who assisted with thoracotomies during rapid autopsies. We thank Dr. Corri Levine, Ms. Nicole Cloutier, and UTMB Biorepository for Severe Emerging Infections for providing the human specimens and data. We thank the research subjects who agreed to participate in this research. We thank Dr. Y. Whitney Yin (Department of Pharmacology and Toxicology, UTMB) for kindly providing the expression clones for purification of nsp 7, nsp 8 and nsp12 structural proteins of SARS-CoV-2 and Dr. Katherine Kaus, Research Development Specialist at UTMB for editing the manuscript.

## Author contributions

T.K.H.; Contributed to conception and design. N.T., A.C., K.S., L.P., K.H., I.M.S., J.D., J.A., V.C., C.T.; Contributed to experimental work. N.T., A.C., K.S.; Contributed to data and statistical analysis, and interpretation of data. D.S., P.G.; Contributed to transcriptome dataset analysis. P.G., N.T. and A.C.; Contributed to data visualization. A.I.; Contributed in purification of recombinant protein. A.H.S., M.W., V.J.C., G.S., L.E.C.A., S.S., S.D., I.B. and G.G.; Contributed with reagents and resources, and provided technical advice. N.T.; Drafted the manuscript, which was revised by T.K.H., I.B., P.G., G.G., S.D. and A.C. All authors read and approved the final manuscript.

## Competing interests

The authors declare no competing interests.

## Additional information

[1]Department of Internal Medicine, University of Texas Medical Branch, Galveston, TX 77555, USA. [2]Department of Chemistry and Biochemistry, University of California, San Diego, La Jolla, CA 92037, USA. [3]Department of Microbiology and Immunology, University of Texas Medical Branch, Galveston, TX 77555, USA. [4]Department of Medicine, Immunology Allergy and Rheumatology, Baylor College of Medicine, Houston, TX 77030, USA. [5]Department of Pathology, University of California, San Diego, CA 92093, USA. [6]Department of Internal Medicine, Division of Cardiology, UC San Diego Medical Center, La Jolla, CA 92037, USA. [7]Department of Cellular and Molecular Medicine, University of California San Diego, La Jolla, CA 92093, USA. [8]Department of Cancer and DNA Damage Responses, Life Sciences Division, Lawrence Berkeley National Laboratory, Berkeley, CA 94720, USA. [9]Department of Neurology, University of Texas Medical Branch, Galveston, TX 77555, USA. [10]Department of Medicine, University of California, San Diego, CA 92093, USA. [11]Department of Pediatrics, University of California San Diego, La Jolla, CA 92093, USA. [12]Department of Computer Science and Engineering, Jacob's School of Engineering, University of California San Diego, La Jolla, CA 92093, USA. [13]Present address: Department of Biological Sciences, School of Engineering and Sciences, SRM University-AP, Guntur District, Andhra Pradesh 522240, India. [14]Present address: Department of Biomedical and Nutritional Science, University of Massachusetts-Lowell, Lowell, MA 01854, USA. [15]These authors contributed equally: Nisha Tapryal, Anirban Chakraborty. ✉e-mail: dsahoo@ucsd.edu; gghosh@ucsd.edu; sodas@ucsd.edu; prghosh@ucsd.edu; sboldogh@utmb.edu; tkhazra@utmb.edu

