## [Peer Review File · Nature Communications]

The DNA glycosylase NEIL2 is protective during SARS-CoV-2 infectionREVIEWER COMMENTS

Reviewer #1 (Remarks to the Author):

By analysing the expression profiles of COVID-19 patients, the authors found that the gene NEIL2, coding for a specific DNA glycosylase, is strongly down-regulated in the lungs. This conclusion was confirmed experimentally that NEIL-2 is decreased by SARS-CoV-2 infection both at the mRNA and protein levels in various cellular models. A humanised mouse model KO for NEIL2 shows a strongest infectivity towards SARS-CoV-2 and a significant increase of severe infections. This phenotype can be reversed by overexpression of NEIL2. By analysing 100 hospitalized COVID-19 patients, they showed that the low levels of NEIL2 are correlated with longer stays at the hospital. In a mouse model *neil2*^{-/-}, the infection by SARS-CoV-2 is significantly more severe and more lethal. Altogether, this data shows that NEIL2 has an overall protective effect against SARS-CoV-2. The authors found that NEIL2 binds specifically to the 5'UTR of SARS-CoV-2 and thereby inhibits viral translation. The conclusion of the manuscript is that the protective effect of NEIL2 against SARS-CoV-2 infection is achieved by specific inhibition of viral translation.

In general, this manuscript brings an impressive amount of novel data on SARS-CoV-2 infection based on several multidisciplinary approaches. The experiments are well designed with appropriate controls and generally well interpreted. However, the part of the manuscript dedicated to the description of NEIL2 binding sites in the 5' and 3'UTR is preliminary and needs to be strengthened. Moreover, the author should confirm the inhibition of viral translation by NEIL2 by analysing its impact on genuine viral proteins such as NSP1 for instance.

Specific points

- In fig 3e, the gel is not annotated, what are the different lanes?
- In fig 5a, the accurate sequence of the SARS-CoV-2 5'UTR used in this experiment should be provided. In addition, in the legend of panel 5b, it should be mentioned that it is the 5'UTR instead of UTR.
- In the EMSA experiments, the used sequences and their localisation in the 5'UTR should be shown. Moreover, the amount of recombinant NEIL2 used in these experiments is from 25 to 100 ng. Rather than the absolute amount, the authors should give the concentration in order to assess the affinity of the protein for the RNA. In these experiments, only a small proportion of the labelled RNA is shifted indicating that the concentration of NEIL2 used is much lower than the dissociation constant K_d. The author should increase the concentration of NEIL2 in order to reach a complete shift of radio-labelled RNA. The authors should also perform an EMSA experiment with the whole 5'UTR of SARS-CoV-2, in this case two binding sites could be detected.
- The same remarks apply to the experiment dedicated to the 3'UTR (Supplementary figure 6b).

Reviewer #2 (Remarks to the Author):

Tapryal et al Nature review

The manuscript by Tapryal et al reports the very unanticipated finding that DNA glycosylase, NEIL2 plays a significant role in modulating the severity of SARS-CoV-2 infection. The progression of these investigations follows a very systematic analysis, in which the experimental design follows a logical progression that led from an insightful observation. The collective group of investigators participating in these studies have considerable expertise in DNA repair and its intersection with control of inflammatory responses, having pioneered the role of OGG1 in airway inflammation.

The origins of this study arose from data mining of publicly-available transcriptomic analyses of individuals with varying levels of disease severity following infection, with transcript levels of NEIL2

showing an impressive relation with the severity of disease in which low level expression equated with the most severe clinical outcomes. This correlation was NEIL2 specific since other DNA glycosylases that are associated with repair of oxidatively-induced DNA damage, including NEIL1, NEIL3, NTH1, and OGG1 showed no relation with disease severity. These results prompted analyses in SARS-CoV-2-infected Syrian hamster lung, with both transcript and enzyme levels decreased and a concomitant increase in DNA damage. Data were also presented for providing an understanding or rationale for differential gender differences with regard to human disease severity. The investigators extended these analyses to measure disease severity in mice that were previously knocked out for Neil2. Consistent with the hypothesis that decreased (or no Neil2) led to more severe disease outcomes, Neil2 knockout mice that were challenged with mouse-adapted CoV-2/MA10 lost weight more rapidly than wild-type counterparts and had a significant increase in morbidity. Inflammation-associated gene expression in Neil2 KO vs WT revealed that 47 of the 84 genes were upregulated in Neil2 KO, but with a significant decrease in the anti-viral IFN-gamma. A corollary strategy was then adopted to increase NEIL2 in the lungs of infected mice by supplementation with recombinant NEIL2. These data demonstrated a strong protective effect of supplementation with NEIL2, but not other glycosylases. Data demonstrating sequence-specific binding to two Zn-finger binding sites in the 5'UTR; inactivation of the Zn-finger binding motif in NEIL2 (C315S) eliminated binding, thus confirming these binding relationships.

Overall, the studies were performed and reported with rigor and consistently justifying the conclusions that were drawn.

Minor recommendations/comments

Having established a molecular mechanism for NEIL2 suppression of the severity of SARS-CoV-2 infection being associated with viral-specific RNA binding, it would be of interest to add to the discussion a contrasting role (or lack thereof) for NEIL2 in other inflammatory-inducing challenges to the lung such as allergen challenge or ozone exposure. Discussion of differences in the source of the inflammation and oxidative challenge in Neil2-deficient vs WT would reinforce the critical role of NEIL2 in this specific pathology.

Since there are multiple Zn-finger proteins that are capable of binding to the 5' and 3' UTR regions, do the authors have comparative binding data or competition assays between NEIL2 and some of these other characterized Zn-finger proteins?

Add a brief discussion of the competing and complementary roles of NEIL2 and OGG1 in regulating cytokine responses.

Do the authors have data on how snps in NEIL2 might affect an individual's susceptibility to infection or whether proteolytic fragments of NEIL2 could diminish viral load and severity of disease.

Reviewer #3 (Remarks to the Author):

The manuscript: The DNA glycosylase NEIL2 plays a vital role in combating SARS-CoV-2 Infection, by Tapryal, et al, provides evidence that individuals with severe SARS-CoV-2-induced disease exhibit reduced expression of the DNA repair enzyme NEIL2, and demonstrates that mice lacking NEIL2 exhibit enhanced susceptibility to SARS-CoV-2 replication and disease. They also provide evidence that NEIL2 interacts with the 5'UTR of SARS-CoV-2 RNA and that NEIL2 may interfere with SARS-CoV-2 protein synthesis. Overall, the manuscript is well written and provides strong evidence that NEIL2 is likely to play an important role in the pathogenesis of SARS-CoV-2-induced disease. This is particularly true to the analysis of patient data and the studies using NEIL2 knockout mice. However, the mechanistic analysis suggesting that NEIL2 acts by binding to the 5' UTR of SARS-CoV-2 and inhibits SARS-CoV-2 translation is dependent on reporters, or non-virus based assays, and validation of these findings in the context of actual SARS-CoV-2 infection is essential for the authors to draw some of the major conclusions.

Major points:

1) The finding that NEIL2 can interact with the SARS-CoV-2 5' UTR is interesting, however these RNA

pulldown experiments do not test whether NEIL2 interacts with the viral 5' UTR in the context of authentic SARS-CoV-2 infection, and instead rely on reporter constructs. Furthermore, the analysis utilizes nuclear extracts (Fig. 5) that are unlikely to be relevant considering that coronaviruses replicate in the cytoplasm. Therefore, the RNA-CHIP analysis should be repeated in the context of virally infected cells to test whether NEIL2 specifically interacts with the 5' UTRs of viral RNAs.

2) Related to point 1, does NEIL2 localize to the cytosol during SARS-CoV-2 infection? If not, how do the authors propose that NEIL2 is interacting with the viral RNA to exert its effects?

3) The effects on the SARS-CoV-2 UTR mediated protein synthesis are interesting, but again solely dependent on reporter constructs. Therefore, this analysis requires validation in the context of SARS-CoV-2 infection, such as pulse chase analysis of viral protein synthesis or polysome analysis in the context of NEIL2 expression.

4) For the studies in Fig. 3, while it was not possible to analyze effects of NEIL2 knockout on DNA damage at day 5 post infection due to the enhanced mortality at this time point in NEIL2^{-/-} mice, it should be possible to measure these effects at earlier time points (e.g. Days 2, 3, or 4 post infection). This is important, since multiple mechanisms are proposed for NEIL2's effects on SARS-CoV-2 disease outcome, and this would provide important information on whether DNA damage levels correlate with the presence or absence of NEIL2.

Our responses to the Reviewers' comments are as follows:

Reviewer 1:

Comment 1: In general, this manuscript brings an impressive amount of novel data on SARS-CoV-2 infection based on several multidisciplinary approaches. The experiments are well designed with appropriate controls and generally well interpreted. However, the part of the manuscript dedicated to the description of NEIL2 binding sites in the 5' and 3'UTR is preliminary and needs to be strengthened. Moreover, the author should confirm the inhibition of viral translation by NEIL2 by analysing its impact on genuine viral proteins such as NSP1 for instance.

Response: We thank the reviewer for the recognition of the novelty and constructive suggestions. To address the concern of the reviewer, we have performed additional experiments showing binding of NEIL2 with full length 5'- and 3'-UTR (Figs. 5b, c; Supplementary Figs. 6b-d) and also, added new data in Fig. 6e showing inhibition of SARS-CoV-2 Spike glycoprotein by transduced recombinant NEIL2 in A549-ACE2 cells.

Specific points

Comment 2: In fig 3e, the gel is not annotated, what are the different lanes?

Response: Fig. 3e is now replaced with Fig. 3f showing DNA damage analysis in *Neil2* KO, in addition to WT mice, at 4 days post infection (dpi) with either mock- or SARS-CoV-2/MA10 virus using a LA-qPCR-based assay. The data show that *Neil2* KO mice indeed accumulated increased levels of DNA damage as compared to their WT counterparts. Proper annotation and detailed information have been included in Fig. 3f and its legend.

Comment 3: In fig 5a, the accurate sequence of the SARS-CoV-2 5'UTR used in this experiment should be provided. In addition, in the legend of panel 5b, it should be mentioned that it is the 5'UTR instead of UTR.

Response: We have now moved the original Figs. 5a and b to the supplement section as Supplementary Figs. 10a and b, respectively, to accommodate the new figures. Per reviewer's suggestions, the full sequence of the SARS-CoV-2 5'-UTR, used in the original Fig. 5a (now Supplementary Fig 10a), has been provided in Supplementary Table 2 and schematically shown in Supplementary Fig. 7a. Also, "UTR" (in the legend of the original Fig. 5b) is now replaced with "5'-UTR" in the legend of the new Supplementary Fig. 10b.

Comment 4: In the EMSA experiments, the used sequences and their localisation in the 5'UTR should be shown. Moreover, the amount of recombinant NEIL2 used in these experiments is from 25 to 100 ng. Rather than the absolute amount, the authors should give the concentration in order to assess the affinity of the protein for the RNA. In these experiments, only a small proportion of the labelled RNA is shifted indicating that the concentration of NEIL2 used is much lower than the dissociation constant K_d . The author should increase the concentration of NEIL2 in order to reach a complete shift of radio-labelled RNA. The authors should also perform an EMSA experiment with the whole 5'UTR of SARS-CoV-2, in this case two binding sites could be detected.

- The same remarks apply to the experiment dedicated to the 3'UTR (Supplementary figure 6b).

Response: We thank the reviewer for the insightful suggestion so that we could further improve the quality of our manuscript. As shown in the revised manuscript, the map of SARS-CoV-2-5'-UTR is now included in Supplementary Fig. 7a and the exact locations of the two short sequences used in EMSA experiments are denoted as colored highlights, along with detailed information provided in the legend. The amount of recombinant NEIL2 and other recombinant proteins are now presented in molar concentrations throughout the manuscript.

Per the reviewer's suggestions, we have performed the titration of NEIL2 with a limited (1nM) amount of RNA probes containing the ZnF sites by RNA-EMSA, and K_d values were determined for each sequence. These data are now presented in Figs. 5e, f and Supplementary Fig. 8. We also performed RNA-EMSA with radiolabeled full length 5'-UTR and 3'-UTRs of SARS-CoV-2 genomic RNA (Sequences are given in Supplementary Table 2), as suggested by the reviewer, and K_d values of NEIL2 for both sequences were also determined. Data are now

presented in **Fig. 5b-d** and **Supplementary Fig. 6b-d**. The detailed discussion regarding the cooperative binding of NEIL2 with SARS-CoV-2 UTRs is provided in the Discussion (paragraph 3).

Reviewer 2:

Comment 1: Overall, the studies were performed and reported with rigor and consistently justifying the conclusions that were drawn.

Response: We appreciate the reviewer for encouraging remarks on the study design.

Minor recommendations/comments

Comment 2: Having established a molecular mechanism for NEIL2 suppression of the severity of SARS-CoV-2 infection being associated with viral-specific RNA binding, it would be of interest to add to the discussion a contrasting role (or lack thereof) for NEIL2 in other inflammatory-inducing challenges to the lung such as allergen challenge or ozone exposure. Discussion of differences in the source of the inflammation and oxidative challenge in Neil2-deficient vs WT would reinforce the critical role of NEIL2 in this specific pathology.

Response: We recently demonstrated that NEIL2 blocks the access of NF-κB, a transcriptional activator of proinflammatory genes, to target gene promoters by directly interacting with the DNA binding domain of RelA (Tapryal et al. JBC. 2021. PMID: 33932404). We found the elevated expression of proinflammatory genes in the *Neil2* KO mice vs. WT mice post SARS-CoV-2 infection, again suggesting the activation of NF-κB signaling in infected *Neil2* KO mice. Thus, we hypothesize that NEIL2 blocks the SARS-CoV-2-infection-induced proinflammatory gene expression by an identical mechanism. Although, we have not tested the anti-inflammatory role of NEIL2 under ozone exposure conditions, we did find increased NF-κB binding at target gene promoters in *Neil2* KO vs. WT mice post allergen challenge (such as ragweed pollen and cat dander extract; unpublished data). Thus, we found the same dominant mechanism for an anti-inflammatory role of NEIL2 under various inflammatory conditions. In the revised manuscript, we have briefly discussed only published results (Discussion, paragraph 2). As some of the data are still unpublished, we opted not to discuss the anti-inflammatory role of NEIL2 under different inflammatory sources in the current manuscript.

Comment 3: Since there are multiple Zn-finger proteins that are capable of binding to the 5' and 3' UTR regions, do the authors have comparative binding data or competition assays between NEIL2 and some of these other characterized Zn-finger proteins?

Response: Multiple host Zn-finger proteins bind to 5'- and 3'-UTRs of viral RNA; however, a considerable knowledge gap exists on whether or exactly how ZnF-containing proteins might protect against SARS-CoV-2 infection. Among these proteins, only zinc finger antiviral protein (ZAP) has been shown to restrict SARS-CoV-2 replication via its interaction with the viral RNA. ZAP recognizes viral RNA via its binding to CpG dinucleotides that are located throughout the viral RNA, and is not just restricted to only 5'- or 3'-UTRs (Afrasiabi et al. Sci Rep. 2022. PMID: 35165300; Ficarelli et al. Annu Rev Virol. 2021. PMID: 34129371). Whereas we

Figure A and B) RNA-EMSA showing the binding of Pol eta (A) or hnRNPU (B) with ³²P-labeled ~38-mer RNA probe containing zinc finger protein (ZnF) binding sites (5'-UTR-ZnF -site-1 or -site-2 and 3'-UTR-ZnF) derived from 5'-UTR or 3'-UTRs of SARS-CoV-2 genomic RNA. Binding of 200 nM NEIL2 is used as positive control.

showed that NEIL2 molecules robustly and preferentially bind to UTRs of SARS-CoV-2, compared to gene body region, as analyzed by RNA-ChIP assays (**Fig. 5a**). Due to completely different mechanisms of viral RNA recognition for ZAP and NEIL2, we did not test binding of ZAP to the SARS-CoV-2 RNA in our experimental system. Nonetheless, we tested the binding of another Zn-finger protein, DNA polymerase eta (Pol eta) that is also involved in DNA repair, and an RNA binding protein hnRNPU with SARS-CoV-2 UTRs by RNA-EMSA. We did not detect any binding of Pol eta or hnRNPU to the SARS-CoV-2 5'- or 3'-UTR regions (See Figures A and B, shown above). Due to lack of direct relevance to the current study, we did not include these data in this revised manuscript. At this stage, it is not feasible to characterize more Zn-finger proteins with respect to their binding activities to SARS-CoV-2 RNA, however, we will pursue this in future studies.

Comment 4: Add a brief discussion of the competing and complementary roles of NEIL2 and OGG1 in regulating cytokine responses.

Response: We have now added a short discussion describing the competing roles of NEIL2 and OGG1 regulating cytokine responses in the proper context (Discussion, paragraph 2).

Comment 5: Do the authors have data on how snps in NEIL2 might affect an individual's susceptibility to infection or whether proteolytic fragments of NEIL2 could diminish viral load and severity of disease.

Response: We appreciate the suggestion of the reviewer. We and others have reported the association of some of the SNPs of NEIL2 with various human pathologies (Dey et al. DNA Repair 2012. PMID: 22497777; Ye et al. Sci Rep. 2020 PMID: 32198476; Broderick et al. BMC Cancer 2006. PMID: 17029639). However, we did not investigate in detail the effect of different SNPs on an individual's susceptibility to SARS-CoV-2 infection. Currently, we do not have data on effect of proteolytic fragments of NEIL2 on viral load and severity of disease. We will definitely pursue these characterizations in the future, however, at this stage, it remains beyond the scope of the present manuscript.

Reviewer 3:

Comment 1: Overall, the manuscript is well written and provides strong evidence that NEIL2 is likely to play an important role in the pathogenesis of SARS-CoV-2-induced disease. This is particularly true to the analysis of patient data and the studies using NEIL2 knockout mice. However, the mechanistic analysis suggesting that NEIL2 acts by binding to the 5' UTR of SARS-CoV-2 and inhibits SARS-CoV-2 translation is dependent on reporters, or non-virus based assays, and validation of these findings in the context of actual SARS-CoV-2 infection is essential for the authors to draw some of the major conclusions.

Response: The authors thank the reviewer for positive remarks and constructive comments. We have addressed the comments raised in the revised manuscript accordingly.

Major points:

Comment 2: The finding that NEIL2 can interact with the SARS-CoV-2 5' UTR is interesting, however these RNA pulldown experiments do not test whether NEIL2 interacts with the viral 5' UTR in the context of authentic SARS-CoV-2 infection, and instead rely on reporter constructs. Furthermore, the analysis utilizes nuclear extracts (Fig. 5) that are unlikely to be relevant considering that coronaviruses replicate in the cytoplasm. Therefore, the RNA-CHIP analysis should be repeated in the context of virally infected cells to test whether NEIL2 specifically interacts with the 5' UTRs of viral RNAs.

Response: The authors apologize for the editorial errors that led to the unnecessary confusion for the reviewer. The RNA-ChIP was indeed performed using whole cell lysate, not nuclear extracts. We have now corrected this in the revised version of the manuscript. Also, the data has been moved to **Supplementary Fig. 10b** to better fit into the scope of the study. Per reviewer's suggestion, we also performed RNA-ChIP using whole cell lysates prepared from human A549-ACE2 cells infected with SARS-CoV-2 and indeed found a robust enrichment of NEIL2 at both 5'- and 3'-UTRs of SARS-CoV-2 genomic RNA, compared to the gene body region, as shown in new **Fig. 5a**.

Comment 3: Related to point 1, does NEIL2 localize to the cytosol during SARS-CoV-2 infection? If not, how do the authors propose that NEIL2 is interacting with the viral RNA to exert its effects?

Response: We have consistently found NEIL2 within cytosolic and nuclear fractions of the cells. We have now included an immunoblot analysis showing sub-cellular localization of NEIL2 by using cytosolic and nuclear extracts from lungs of uninfected mice (**Supplementary Fig. 6a**). Immunoblot analyses with GAPDH and HDAC2 clearly confirm the purity of the two fractions.

Comment 4: The effects on the SARS-CoV-2 UTR mediated protein synthesis are interesting, but again solely dependent on reporter constructs. Therefore, this analysis requires validation in the context of SARS-CoV-2 infection, such as pulse chase analysis of viral protein synthesis or polysome analysis in the context of NEIL2 expression.

Response: At this stage, we are unable to perform pulse chase or polysome analysis of viral protein in the context of NEIL2 expression due to technical constraints. However, we performed immunoblot analysis of SARS-CoV-2 Spike glycoprotein in cell lysates from *Neil2* KO vs. WT mice lungs post SARS-CoV-2/MA10 infection and found significantly higher levels of Spike protein in *Neil2* KO samples, suggesting that NEIL2 modulates viral protein levels (**Fig. 3e**, lanes 8-10 vs. 5-7). Furthermore, we performed immunoblot analysis for Spike glycoprotein in lysates from A549-ACE2 cells transduced with recombinant NEIL2 protein prior to SARS-CoV-2 infection, and found a significantly lower level of Spike protein in NEIL2 transduced cells at 24 and 48 h post infection (**Fig. 6e**). These data, in combination with the assays using reporter constructs, confirm the inhibitory role of NEIL2 in viral protein synthesis.

Comment 5: For the studies in Fig. 3, while it was not possible to analyze effects of NEIL2 knockout on DNA damage at day 5 post infection due to the enhanced mortality at this time point in NEIL2^{-/-} mice, it should be possible to measure these effects at earlier time points (e.g. Days 2, 3, or 4 post infection). This is important, since multiple mechanisms are proposed for NEIL2's effects on SARS-CoV-2 disease outcome, and this would provide important information on whether DNA damage levels correlate with the presence or absence of NEIL2.

Response: We have now conducted DNA damage analysis with mock or SARS-CoV-2/MA10 infected *Neil2* KO vs. WT mice lungs at 4 dpi and found that *Neil2* KO mice accumulate significantly higher levels of DNA strand breaks compared to WT mice post SARS-CoV-2 infection (**Fig. 3f**). Thus, the absence of NEIL2 indeed correlates well with higher DNA damage associated with COVID-19 pathogenesis, and greatly affects the SARS-CoV-2 disease outcome as proposed.

REVIEWERS' COMMENTS

Reviewer #1 (Remarks to the Author):

In the revised version of their manuscript, the authors have carefully addressed my concerns about the binding sites of NEIL-2 on the SARS-CoV-2 5' and 3'UTR and performed the requested experiments.

Reviewer #2 (Remarks to the Author):

The authors have done an outstanding job being responsive to the initial set of reviews, including additional analyses, modifying text, expanding the discussion of important points, and appropriately deferring some suggestions to future investigations.
No additional modifications are needed at this time.

Reviewer #2

R. Stephen Lloyd

Reviewer #3 (Remarks to the Author):

The authors have appropriately addressed my prior concerns.

REVIEWERS' COMMENTS

Reviewer #3 (Remarks to the Author):

This is a revised version of the manuscript "The DNA glycosylase NEIL2 plays a vital role in combating SARS-CoV-2 infection," by Tapryal, et al, identifies the DNA repair gene, NEIL2, as a modulator of SARS-CoV2 pathogenesis. The studies, which build on findings that NEIL2 expression levels correlate with severity of COVID-19 disease, provide evidence that NEIL2 modulates virus-induced inflammation, while also serving an antiviral role through binding to viral UTRs and modulating the translation of viral proteins.

The revised manuscript is significantly improved both through the inclusion of new scientific data, and the authors have appropriately addressed my concerns from the previous review.

Our responses to the Reviewers' comments are as follows:

Reviewer #3 (Remarks to the Author):

This is a revised version of the manuscript " The DNA glycosylase NEIL2 plays a vital role in combating SARS-CoV-2 infection," by Tapryal, et al, identifies the DNA repair gene, NEIL2, as a modulator of SARS-CoV2 pathogenesis. The studies, which build on findings that NEIL2 expression levels correlate with severity of COVID-19 disease, provide evidence that NEIL2 modulates virus-induced inflammation, while also serving an antiviral role through binding to viral UTRs and modulating the translation of viral proteins.

The revised manuscript is significantly improved both through the inclusion of new scientific data, and the authors have appropriately addressed my concerns from the previous review.

Response: We thank the reviewer for the recognition of the novelty and impact of the current study. We appreciate reviewer's constructive comments that helped us improve the study.